# Functional annotation of enzyme-encoding genes using deep learning with transformer layers

Gi Bae Kim [1,2,3], Ji Yeon Kim[1,2,3], Jong An Lee[1,2,3], Charles J. Norsigian[4,5], Bernhard O. Palsson [5,6,7] & Sang Yup Lee [1,2,3,8] ✉

Functional annotation of open reading frames in microbial genomes remains substantially incomplete. Enzymes constitute the most prevalent functional gene class in microbial genomes and can be described by their specific catalytic functions using the Enzyme Commission (EC) number. Consequently, the ability to predict EC numbers could substantially reduce the number of un-annotated genes. Here we present a deep learning model, DeepECtransformer, which utilizes transformer layers as a neural network architecture to predict EC numbers. Using the extensively studied *Escherichia coli* K-12 MG1655 genome, DeepECtransformer predicted EC numbers for 464 un-annotated genes. We experimentally validated the enzymatic activities predicted for three proteins (YgfF, YciO, and YjdM). Further examination of the neural network's reasoning process revealed that the trained neural network relies on functional motifs of enzymes to predict EC numbers. Thus, DeepECtransformer is a method that facilitates the functional annotation of uncharacterized genes.

Enzymes are proteins that catalyze various reactions in living organisms. Understanding the functions of enzymes is vital for comprehending metabolic processes and characteristics. The International Union of Biochemistry and Molecular Biology devised the Enzyme Commission (EC) number system, which assigns enzyme functions using four hierarchical digits separated by periods (e.g., EC:1.1.1.1 for alcohol dehydrogenase). EC numbers can facilitate the annotation and classification of enzymes in the growing number of genome sequences, playing a crucial role in understanding the metabolism of organisms and enabling metabolic engineering applications.

The surge in biological sequence data, along with the advancement of data-driven methodologies, particularly deep learning, has facilitated the large-scale characterization of proteins[1–10]. Deep learning has proven useful in analyzing biological sequences, including enzyme functions[4,11,12], turnover numbers[13–15], and Michaelis constants[16]. Although deep learning models have been criticized for being "black boxes," recent studies have used various methods, such as integrated gradients, to interpret the reasoning process of artificial intelligence (AI)[17–22]. For instance, we used the integrated gradients method to interpret the reasoning process of DeepTFactor, a deep learning tool for transcription factor prediction, demonstrating that AI comprehends DNA-binding domains of transcription factors without having been trained with such information[20]. Interpreting deep learning models is expected to deepen our understanding of the models' processes and unveil unknown biological features.

[1]Metabolic and Biomolecular Engineering National Research Laboratory, Department of Chemical and Biomolecular Engineering (BK21 four), Korea Advanced Institute of Science and Technology (KAIST), Daejeon 34141, Republic of Korea. [2]Systems Metabolic Engineering and Systems Healthcare Cross-Generation Collaborative Laboratory, Department of Chemical and Biomolecular Engineering (BK21 four), KAIST, Daejeon 34141, Republic of Korea. [3]KAIST Institute for the BioCentury and KAIST Institute for Artificial Intelligence, KAIST, Daejeon 34141, Republic of Korea. [4]Division of Biological Sciences, University of California San Diego, La Jolla, CA 92093, USA. [5]Department of Bioengineering, University of California San Diego, La Jolla, CA 92093, USA. [6]Bioinformatics and Systems Biology Program, University of California San Diego, La Jolla, CA 92093, USA. [7]Novo Nordisk Foundation Center for Biosustainability, 2800 Kongens Lyngby, Denmark. [8]BioProcess Engineering Research Center and BioInformatics Research Center, KAIST, Daejeon 34141, Republic of Korea. ✉e-mail: leesy@kaist.ac.kr

Various deep learning models for the prediction of EC numbers have also been developed. For instance, HDMLF was developed by integrating multiple sequence alignment with a deep neural network leveraging learned representations from a protein language model and bidirectional gated recurrent units[23]. CLEAN, another deep learning model addressed imbalances in EC number distribution within the training dataset by employing contrastive learning, leading to prediction performance superior to the previous models[24]. However, these models did not provide insights into the interpretability of AI reasoning. ProteInfer used a deep dilated convolutional network for EC number prediction and also provided interpretation of the prediction by class activation mapping[25]. Nevertheless, the class activation mapping yielded coarse-grained feature maps, lacking fine-grained details, which are important for the residue-level analysis of protein sequences. We previously developed DeepEC, a deep learning-based computational framework for EC number prediction that uses only the amino acid sequences of proteins as input[4] (Supplementary Fig. 10). In this study, we present the development of DeepECtransformer, which employs transformer layers as the neural network architecture to effectively predict EC numbers, covering 5360 EC numbers and including the EC:7 class (translocase) that was previously not covered in DeepEC. Also, the improved performance of DeepECtransformer has suggested a list of entries that need careful inspection of whether they have mis-annotated EC numbers in the UniProt Knowledgebase (UniProtKB).

By analyzing the regions of focus during the prediction of enzyme functions by transformer layers, we have confirmed that DeepECtransformer has learned to identify important regions, such as active sites or cofactor binding sites. To unveil the functions of y-ome (unknown) proteins in the model organism *Escherichia coli* K-12 MG1655, we employed DeepECtransformer to predict EC numbers for 464 proteins out of 1569 y-ome proteins. Out of the 464 proteins, the functions of three predicted enzymes (YgfF, YciO, and YjdM) were validated through in vitro enzyme activity assays. This demonstrates the capability of DeepECtransformer not only to quickly annotate enzyme functions from increasing amounts of DNA sequences but also to discover metabolic functions of proteins that were previously unknown.

## Results
### Development and evaluation of DeepECtransformer
DeepECtransformer incorporates two prediction engines: a neural network and a homologous search. The neural network uses a transformer architecture to predict EC numbers by extracting latent features from the amino acid sequences of enzymes (Fig. 1a)[8,26]. The neural network was trained on a uniprot dataset consisting of the amino acid sequences of 22 million enzymes from UniProtKB/TrEMBL entries, covering 2802 EC numbers with all four digits (see "Methods")[27]. If the neural network predicts no EC numbers for a given amino acid sequence, homologous enzymes for the amino acid sequence are analyzed using UniProtKB/Swiss-Prot enzymes as the subject database and EC numbers of the homologous enzymes are assigned[4,28]. Including EC numbers that can be predicted by neural network and homology search, DeepECtransformer covers a total of 5360 EC numbers (Supplementary Fig. 11).

The performance of the neural network was evaluated on a separate test dataset that was not used during training. The performance results varied depending on the EC number class, with precision ranging from 0.7589 to 0.9506, recall ranging from 0.6830 to 0.9445, and $F_1$ score ranging from 0.6990 to 0.9469 (Fig. 1b). The evaluation metrics (precision, recall, and $F_1$ score) of the neural network were lowest for the EC:1 class (oxidoreductases), which made up 13.4% of enzymes (i.e., 313,328 sequences) and made up 25.7% of EC numbers (i.e., 720 EC numbers) in the uniprot dataset (Fig. 1c, d). The low performance for EC:1 class resulted from the inherent dataset

imbalance, as the EC:1 class exhibited the lowest average number of sequences per EC number, with an average of 4352 sequences compared to the other EC number classes (ranging from 6819 sequences for EC:3 to 16,525 sequences for EC:6). Additionally, a statistical analysis of the data distribution confirmed that EC numbers belonging to EC:1 class generally had fewer sequences compared to other EC number classes (one-way ANOVA test, $p$ value < 7.2473e−15) (Supplementary Fig. 1). Hence, the neural network showed a bias towards varying performances across different EC numbers. Furthermore, a positive correlation was observed between the $F_1$ score and the number of sequences per EC number (Spearman coefficient of 0.6872, $p < 0.001$, $n = 2802$; Fig. 1e).

The performance of DeepECtransformer was evaluated by comparing it with two baseline methods: DeepEC and a homology-based search tool, DIAMOND[4,28]. For the comparison, the test dataset that has been used for the evaluation of DeepECtransformer neural network was curated to consist of the amino acid sequences of 2,013,612 enzymes, for which EC numbers can be predicted by all three tools. DeepECtransformer showed superior performance in terms of precision, recall, and $F_1$ score, with the exception of micro precision, which was slightly lower than those of DIAMOND and DeepEC (Table 1 and Supplementary Fig. 2). Moreover, DeepECtransformer demonstrated an improved ability to predict EC numbers for enzymes that have low sequence identities to those in the training dataset (Supplementary Fig. 3).

The accuracy of DeepECtransformer was further demonstrated by its ability to correct mis-annotated EC numbers in UniProtKB. An example is the enzyme P93052 from *Botryococcus braunii*, which was originally annotated as an L-lactate dehydrogenase (EC:1.1.1.27)[29]. However, DeepECtransformer predicted it as a malate dehydrogenase (EC:1.1.1.37). We performed heterologous expression experiments (Supplementary Fig. 12 and Supplementary Notes 1 and 4), which confirmed that P93052 is a malate dehydrogenase (EC:1.1.1.37). Similarly, DeepECtransformer correctly predicted EC numbers for Q8U4R3 from *Pyrococcus furiosus* and Q038Z3 from *Lacticaseibacillus paracasei* as D-cysteine desulfhydrase (EC:4.4.1.15) and dihydroorotate dehydrogenase (NAD) (EC:1.3.1.14), respectively, which were misannotated as 1-aminocyclopropane-1-carboxylate deaminase (EC:3.5.99.7) and dihydroorotate dehydrogenase (fumarate) (EC:1.3.98.1). DeepECtransformer made predictions for 26,140 proteins with EC numbers that differed from those in Swiss-Prot (Supplementary Data 1). Among them, 2062 proteins were predicted to have additional EC numbers. For example, Q9WVK7 was annotated as EC:1.1.1.35, NAD-dependent 3-hydroxyacyl-CoA dehydrogenase in Swiss-Prot, while DeepECtransformer predicted it as EC:1.1.1.157, NADP-dependent 3-hydroxybutyryl-CoA dehydrogenase, in addition to EC:1.1.1.35. Additionally, DeepECtransformer could fill in the incomplete EC numbers of 6454 proteins. It added the fourth digit to 5012 proteins, the third and fourth digits to 1019 proteins, and the second, third, and fourth digits to 423 proteins. For example, DeepECtransformer predicted C9K7D8 as EC:2.6.1.42, a branched-chain amino acid transaminase, which was previously annotated as EC:2.6.1, a transaminase. As the EC numbers suggested by DeepECtransformer are predictions, we further analyzed how AI made the predictions in silico (Supplementary Note 2 and Supplementary Figs. 14–17). Even though in silico analysis of the predictions can provide clues for the predicted functionality, it is necessary to experimentally validate their functions. However, given the rapidly increasing numbers of genomes and metagenomes, it is not feasible to experimentally validate the functions of all unknown proteins. The predictions made by DeepECtransformer can provide candidate entries for further review, contributing to the construction of a more robust knowledgebase.

### AI learns the functional regions of enzymes
DeepECtransformer can classify enzymes based on their EC numbers by utilizing inherent filters that extract latent features from the amino acid sequences of enzymes. To understand the classification

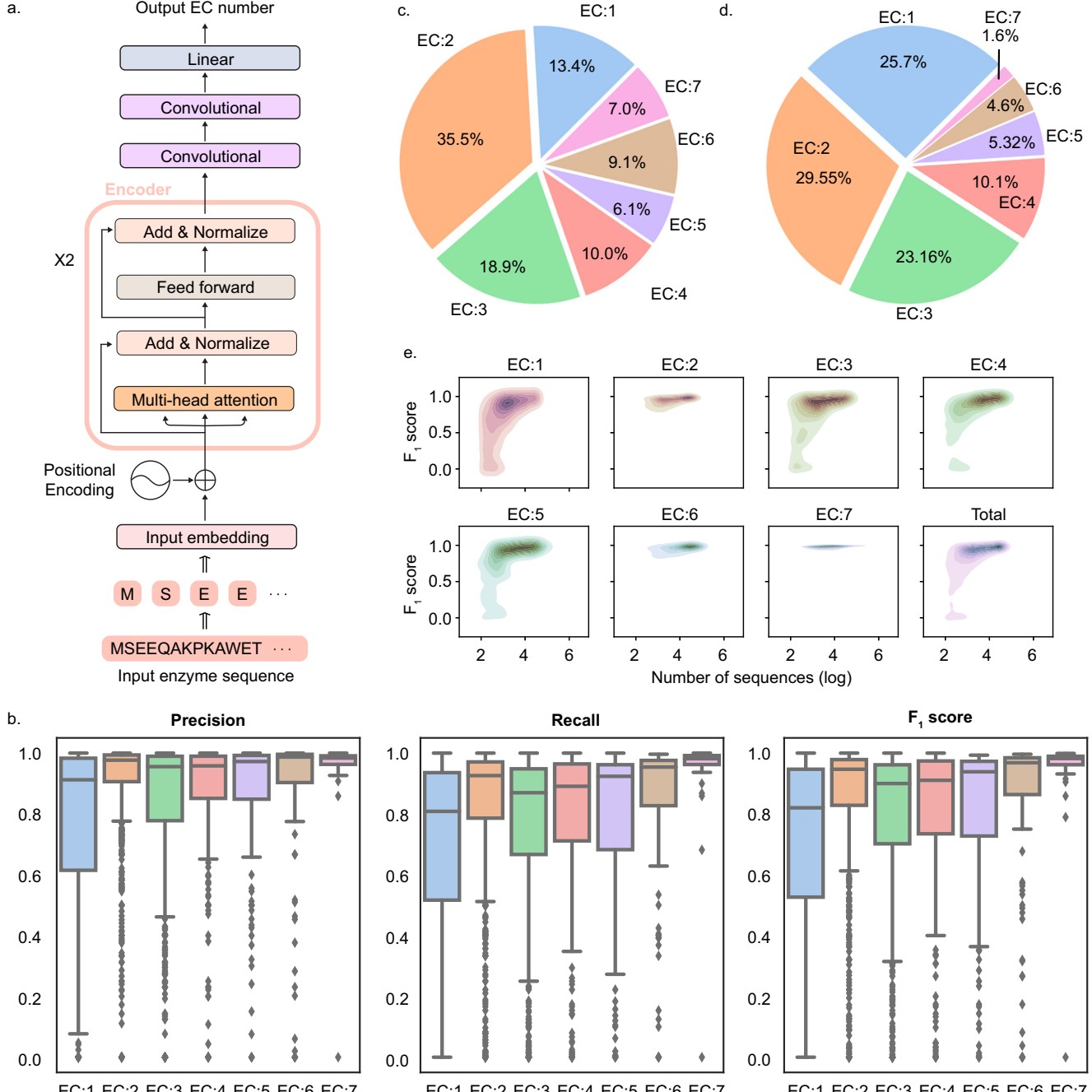

**Fig. 1 | The network architecture of DeepECtransformer and the prediction performance of the neural network. a** Network architecture of DeepECtransformer. DeepECtransformer employs the BERT architecture adopted from ProtTrans[26,43]. The network consists of two transformer encoders, two convolutional layers, and a linear layer. Taking the amino acid sequence of an enzyme, the neural network predicts the EC numbers of the enzyme. **b** The prediction performance of the neural network for the test dataset. The boxplots illustrate the varying performance of the neural network by the first-level EC numbers. Each data point in the boxplot represents the performance of the neural network for a single EC number. The precision, recall, and $F_1$ score differ for EC number classes. Center line, box limits, whiskers, and points of the box-plots represent median, upper and lower quartiles, 1.5× interquartile range, and outliers, respectively. The sample sizes of the boxplots for EC:1, EC:2, EC:3, EC:4, EC:5, EC:6, and EC:7 are 720, 828, 649, 283, 149, 129, and 44, respectively. **c** Distribution of the amino acid sequences of enzymes per first-level EC number. **d** Distribution of types of EC numbers per first-level EC number. The amino acid sequences of enzymes in the uniprot dataset were used to analyze the distributions. **e** The prediction performance of the neural network by the number of the amino acid sequences of enzymes in the uniprot dataset. Source data are provided as a Source data file.

mechanism, we projected the latent vectors of the amino acid sequences into a two-dimensional space using TMAP (Supplementary Data 2)[30]. Although enzymes with identical EC numbers tend to cluster together, this pattern is less prominent at higher levels of EC classification (e.g., EC numbers with second digit). This suggests that the neural network may not consider the common features of higher-level

EC classes during the classification process. To examine the specific features learned by the network, we analyzed the attention scores computed in the self-attention layers (see "Methods"). For instance, DeepECtransformer uses the active site and NAD binding residues to predict the EC number of NAD-dependent malate dehydrogenase (EC:1.1.1.37) in *E. coli* K-12 (Fig. 2a and Supplementary Fig. 4). Similarly,

**Table 1 | Performance of EC number prediction tools for the test dataset**

| Tool | Number of predicted sequences | Macro precision | Macro recall | Macro $F_1$ score | Micro precision | Micro recall | Micro $F_1$ score |
|------|------|------|------|------|------|------|------|
| DIAMOND | 1,300,039 | 0.8168 | 0.4590 | 0.5390 | 0.9813 | 0.5867 | 0.7343 |
| DeepEC | 1,272,079 | 0.8357 | 0.3931 | 0.4802 | 0.9727 | 0.6062 | 0.7469 |
| DeepECtransformer | 1,952,172 | 0.8537 | 0.7942 | 0.8093 | 0.9709 | 0.9516 | 0.9611 |

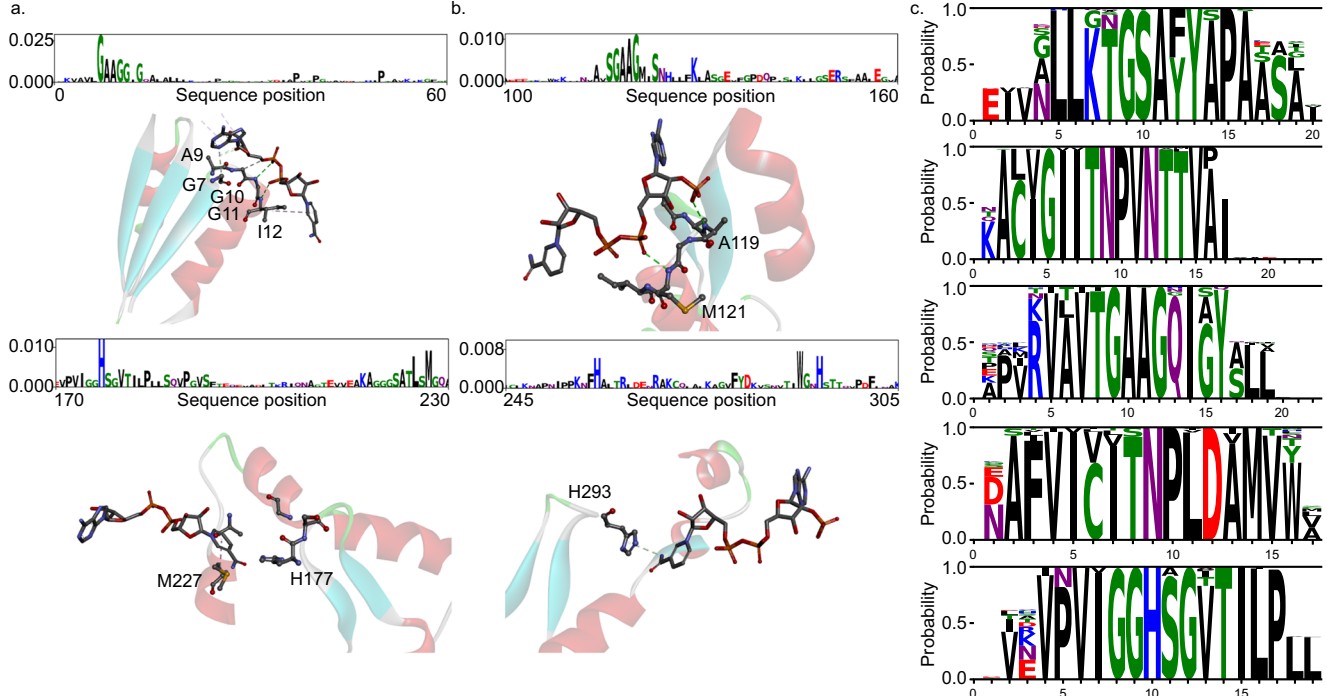

**Fig. 2 | Highlighted amino acid residues by the DeepECtransformer neural network. a** DeepECtransformer pays attention to the NAD binding residues and the active site to predict the EC number of NAD-dependent malate dehydrogenase of *E. coli* K-12 (Protein Data Bank [PDB] ID code 1IB6). **b** DeepECtransformer pays attention to the NADP binding residues and the active site to predict the EC number of NADP-dependent malate dehydrogenase of *F. bidentis* (PDB ID code 1CIV). The whole attention scores of the neural network prediction for the enzymes are available in Supplementary. 4 and Supplementary Fig. 5. **c** Commonly highlighted motifs for prediction of NAD-dependent malate dehydrogenase by DeepEC-transformer neural network.

in the prediction of the EC number of NADP-dependent malate dehy-drogenase (EC:1.1.1.82) in *Flaveria bidentis*, DeepECtransformer assigned high attention scores to the active site and NADP binding residues (Fig. 2b and Supplementary Fig. 5). These results suggest that DeepECtransformer can identify the critical motifs in the amino acid sequences of enzymes without requiring any prior knowledge. To systematically examine the functional motifs utilized by DeepEC-transformer, the common motifs that contain residues with high attention scores in the self-attention layer were extracted. Attention scores were computed for the residues in the amino acid sequences of enzymes in Swiss-Prot for each EC number. The motif with the highest attention score in each attention head was extracted and clustered to identify the common motifs. Multiple sequence alignments were conducted for each cluster, which revealed up to five common motifs for each EC number that were assigned high scores by DeepEC-transformer. These common motifs represent functionally important residues, such as active sites or substrate binding sites. For example, the common motifs for NAD-dependent malate dehydrogenase (EC:1.1.1.37) were identified to comprise two Pfam domains (PF00056 and PF02866) that correspond to the NAD binding domain and α/β C-terminal domain of malate dehydrogenase, respectively (Fig. 2c). Similarly, the common motifs for pyruvate kinase (EC:2.7.1.40) include a Pfam domain PF00224, which corresponds to the pyruvate kinase barrel domain. Additionally, the common motifs for fumarase

(EC:4.2.1.2) contain a Pfam domain PF10415, which corresponds to the fumarase C C-terminus. To provide functional units for EC numbers that can be predicted by the DeepECtransformer neural network, we extracted commonly highlighted motifs for 2547 EC numbers from the amino acid sequences of enzymes in Swiss-Prot (Supplementary Data 3). These motifs are expected to be utilized in future studies to uncover unknown shared characteristics among proteins with corre-sponding EC numbers.

**Analysis of the metabolic function of alleles for *E. coli* strains**
To evaluate the ability of DeepECtransformer in predicting changes in metabolic functions across different strains, DeepECtransformer and DIAMOND were employed to predict the EC numbers of 312,274 pro-teins encoded by 3967 genes in 1122 *E. coli* strains collected from the NCBI Genome database (Supplementary Data 4). The metabolic func-tions of the enzymes encoded by different alleles may vary although they are annotated as the same gene. Out of the 312,274 proteins, 238,575 proteins exhibited identical predictions from both DeepEC-transformer and DIAMOND. For 2732 genes, representing 68.87% of the total, at least 90% of the alleles showed identical predictions between DeepECtransformer and DIAMOND (Fig. 3a). Among the 73,669 alleles with non-identical predictions, 41,372 were newly pre-dicted as enzymes, 1270 were predicted to lose their metabolic func-tions, and 31,057 were predicted to undergo changes in their metabolic

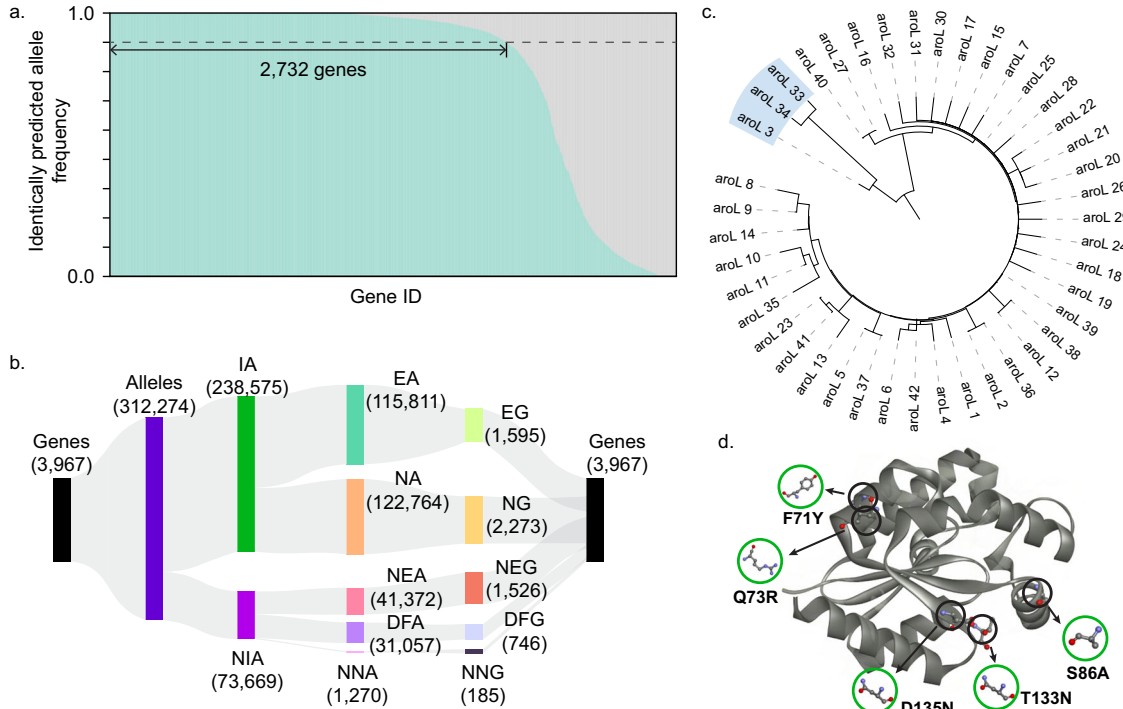

**Fig. 3 | EC number prediction results for 312,274 alleles in 1,122 *E. coli* strains.** **a** Frequency of alleles identically predicted by DeepECtransformer and DIAMOND. **b** Sankey diagram of the number of genes and alleles by the EC number prediction. Numbers in the parentheses indicate the number of each item. Abbreviations are as follows: IA identically predicted alleles, NIA non-identically predicted alleles, EA alleles for enzymes, NA alleles for non-enzymes, NEA alleles newly predicted as enzymes, DFA alleles predicted to differ in functions, NNA alleles newly predicted as non-enzymes, EG genes for enzymes, NG genes for non-enzymes, NEG genes for newly predicted as enzymes, DFG gene for predicted to differ in functions, NNG gene for newly predicted as non-enzymes. **c** The phylogenetic tree of *aroL* alleles in 1122 *E. coli* strains. Three alleles (*aroL*_3, *aroL*_33, and *aroL*_34) are located in a distinct clade. **d** The three-dimensional structure of AroL and common mutations in *aroL*_3, *aroL*_33, and *aroL*_34 (AlphaFoldDB ID code: P0A6E1). Source data are provided as a Source data file.

functions (Fig. 3b). Through the analysis of these non-identically pre-dicted alleles, we were able to analyze how mutations in the alleles affected their metabolic functions.

For instance, among the 42 alleles of the *aroL* gene that encodes shikimate kinase II (EC:2.7.1.71), three alleles (*aroL*_3, *aroL*_33, and *aroL*_34) were predicted to have an additional metabolic function of 7-α-hydroxysteroid dehydrogenase (EC:1.1.1.159). In the phylogenetic analysis of *aroL* alleles, theses alleles (*aroL*_3, *aroL*_33, and *aroL*_34) formed a distinct clade in the phylogenetic tree, suggesting a diver-gent evolutionary trajectory compared to other alleles (Fig. 3c, d). The strains possessing these three alleles (i.e., *E. coli* KTE11, KTE31, KTE33, KTE96, KTE114, and KTE159) exhibit a common feature of carrying the allele *hdhA*_61, which encodes 7-α-hydroxysteroid dehydrogenase. In a recent study, these six strains were also identi-fied as belonging to the same phylogroup of *E. coli* strains[31]. Further investigation into the coevolutionary relationship between *hdhA*_61 and the three *aroL* alleles may provide insights into how theses strains have adapted to their environment. There are other examples of changes in metabolic functions found among the alleles within the distinct clades of phylogenetic trees. For instance, the *lsrF* gene (encoding 3-hydroxy-5-phosphooxypentane-2,4-dione thiolase; EC:2.3.1.245) has six alleles predicted to have an additional metabolic function of fructose-bisphosphate aldolase (EC:4.1.2.13), while *ttdA* (encoding L-tartrate dehydratase; EC:4.2.1.32) has three alleles pre-dicted to have an additional metabolic function of fumarase (EC:4.2.1.2) (Supplementary Fig. 6). Such observations can provide valuable insights into the evolutionary trajectories of strains from a metabolic perspective. In summary, these findings demonstrate that DeepECtransformer can detect changes in metabolic functions resulting from only a few mutations, which are not easily identifiable through homologous searches.

## Discovering the unknown functions of enzymes in *E. coli* K-12 MG1655

As discussed earlier, DeepECtransformer has shown the capability to predict enzyme functions for proteins with low sequence identities to enzymes seen by the model during the training. Therefore, our next objective was to employ DeepECtransformer to uncover unknown metabolic functions of enzymes. *E. coli* K-12 MG1655, an extensively studied model organism, still has approximately 30% of genes that remain incompletely characterized. Utilizing DeepECtransformer, we conducted an analysis of the EC numbers associated with *E. coli*'s y-ome, which comprises genes in *E. coli* K-12 MG1655 with insufficient experi-mental evidence regarding their functions. Out of the 1600 genes in the y-ome, protein sequences for 1569 were retrievable from the UniProt database. DeepECtransformer successfully predicted EC numbers for 464 proteins, with 390 of them having complete four-digit EC numbers (Fig. 4a and Supplementary Data 5). In comparison, our previous algo-rithm DeepEC predicted EC numbers for 82 proteins, of which 71 were projected to possess complete four-digit EC numbers, while the UniProt database provided annotations for 71 of these proteins (Supplementary Fig. 7). DeepECtransformer exclusively predicted complete four-digit EC numbers for 295 proteins.

To validate the ability of DeepECtransformer in identifying meta-bolic functions that cannot be detected by DeepEC and Swiss-Prot functional annotation processes, we performed in vitro enzyme assays to validate the predicted enzyme functions. We first selected three representative EC number classes, namely oxidoreductase (EC:1), transferase (EC:2), and hydrolase (EC:3) to show the capability of DeepECtransformer to predict unknown enzyme functions. Among the 295 proteins that are exclusively predicted by DeepEC-transformer to have all four digits of EC numbers, 179 proteins are predicted to be soluble in *E. coli* by NetSolP, a deep learning model for

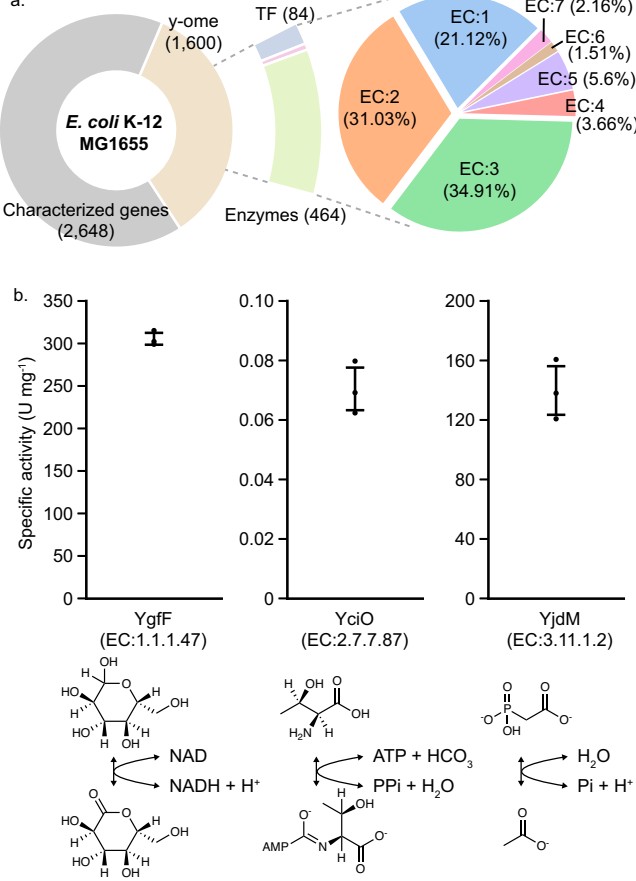

**Fig. 4 | EC number prediction results for proteins of the *E. coli* K−12 MG1655 y-ome. a** Distribution of predicted EC numbers of the y-ome proteins. The number of the transcription factor (TF) is predicted using DeepTFactor. **b** Specific activities of YfgF, YciO, and YjdM. Error bars, mean ± SD (*n* = 3 independent experiments). Black circles represent each data point. Source data are provided as a Source data file.

protein solubility prediction[32]. From the 179 proteins, we randomly selected three proteins, YgfF, YciO, and YdjM, that are predicted to be oxidoreductase, transferase, and hydrolase, respectively. For YgfF, DeepECtransformer predicted its EC number to be EC:1.1.1.47 (glucose 1-dehydrogenase). The enzyme assay results showed that YgfF exhibited a specific glucose 1-dehydrogenase activity of 305.55 U mg$^{-1}$ (Fig. 4b and Supplementary Table 1), which was comparable to the previously reported value of 205.70 U mg$^{-1}$ for the glucose 1-dehydrogenase from *Lysinibacillus sphaericus* G10[33]. In the case of YciO, which was previously annotated to belong to the SUA5 family, DeepECtransformer predicted its EC number to be EC:2.7.7.87 (L-threonylcarbamoyladenylate synthase) with the prediction score of 0.9108. The specific activity of YciO was measured to be 0.0705 U mg$^{-1}$ (Fig. 4b and Supplementary Table 1), which was higher than the previously reported specific activity range of L-threonylcarbamoyladenylate synthase (0.00556–0.01112 U mg$^{-1}$) from *Bacillus subtilis*[34]. Lastly, YjdM was predicted by DeepECtransformer to have the EC number EC:3.11.1.2 (phosphonoacetate hydrolase). The specific phosphonoacetate hydrolase activity of YjdM obtained by enzyme assay was 139.85 U mg$^{-1}$ (Fig. 4b and Supplementary Table 1), which exceeded the specific activity of phosphonoacetate hydrolase (60 U mg$^{-1}$) from *Pseudomonas fluorescens* 23F[35]. We also analyzed whether the EC number predictions for these three enzymes were made by the neural network or by the homology search. First, YgfF was predicted to have an EC number of EC:1.1.1.47 by the neural network with a prediction score of 0.6331. It should be noted that although YgfF exhibited a higher sequence identity with a different enzyme

(A0A069CGU9_ECOLX; EC:1.1.1.100) from the training dataset than glucose 1-dehydrogenase exhibiting the maximum sequence identity within the training dataset, the neural network made an accurate prediction. Likewise, for YjdM, predicted by the neural network as EC:3.11.1.2 with a prediction score of 0.6103, the training sequence with the highest sequence similarity had a different EC number (C9Y1B8_CROTZ; EC:2.7.7.6). Lastly, the predicted EC number for YciO by the neural network was EC:3.1.11.2 with a prediction score of 0.9108, and the training sequence with the highest sequence identity with YciO was also a L-threonylcarbamoyladenylate synthase (EC:3.1.11.2). To examine whether the neural network understands the functional regions of YciO rather than relying on the sequence identity, the motifs with high attention scores were analyzed. It was found that the highlighted motifs correspond to TIGR00057, a NCBIfam family for L-threonylcarbamoyladenylate synthase (Supplementary Fig. 13). These results suggest that DeepECtransformer leverages not only homology search but employs latent features learned during the training process during the prediction of EC numbers. Notably, DeepEC was unable to provide predictions for the three proteins, likely due to its low recall associated with predicting enzyme functions, stemming from a highly imbalanced dataset. Also, Swiss-Prot functional annotation failed to make predictions for the three proteins, possibly due to the limited sequence identity shared between the y-ome proteins and protein signatures available in existing databases. These results suggest that DeepECtransformer can effectively contribute to the discovery of metabolic functions of enzymes that have yet to be fully characterized.

## Discussion

EC numbers are a four-digit code that describes the catalytic function of an enzyme. We developed DeepECtransformer, which combines deep learning with transformer layers and homologous searches, to predict EC numbers. DeepECtransformer was found to outperform both DeepEC and homologous search using DIAMOND in predicting four-digit EC numbers. The neural network of DeepECtransformer was trained using the amino acid sequences of 22 million enzymes covering 2802 EC numbers with all four digits. Despite the large number of sequences in the dataset, the class imbalance made it challenging to predict EC numbers that had few sequences. To address this issue, we trained DeepECtransformer using focal loss, which reduces the impact of class imbalance during training (see "Methods")[36]. Using weighted loss functions[37] or creating a more balanced dataset through data augmentation[38] can further enhance the prediction performance of the neural network.

There have been other attempts to create high-performance neural networks for biological sequences, such as splitting datasets based on sequence identities[10,39], using learned protein representations[39], and incorporating evolutionary information[40,41]. A recently developed EC number prediction tool, CLEAN, has used contrastive learning to address class imbalances by aiming to differentiate data classes in the learned latent space, resulting in improved performance compared to DeepECtransformer (Supplementary Note 3 and Supplementary Tables 2–4)[24]. However, while CLEAN showed improved performance compared to DeepECtransformer, it was not able to predict the EC numbers of YgfF and YjdM. Also, the use of CLEAN for the annotation of uncharacterized proteins requires careful interpretation of the prediction results because CLEAN assigns EC numbers for any input amino acid sequences, including non-enzyme amino acid sequences. CLEAN provides the confidence level (i.e., high, medium, low) of the predictions using a Gaussian mixture model. However, as the confidence level does not provide a detailed interpretation of how AI performs the reasoning process, careful inspection of the predictions should be conducted for the analysis of uncharacterized proteins, especially when the uncharacterized proteins contain non-enzyme proteins. While DeepECtransformer has an advantage in terms of interpretability by providing attention weight-based fine-grained details, it is important to

acknowledge that other EC number prediction tools address different facets of the problem. For instance, CLEAN tackles class imbalance by utilizing contrastive learning rather than supervised learning. ProteInfer uses a deep dilated convolutional network, which is not constrained by input amino acid sequence length and extends its predictions to include Gene Ontology (GO) terms, thereby offering a richer information of protein functionality. This highlights the importance of considering an ensemble of prediction tools for comprehensive enzyme function characterization. Further optimization of the neural network could lead to the development of an AI tool for more accurate enzyme function characterization.

Here, we used DeepECtransformer to predict EC numbers for the y-ome of the well-studied *E. coli* K-12 MG1655 genome to reveal putative enzymatic functions for 464 proteins. We experimentally validated the predicted EC numbers for three enzymes, YgfF, YciO, and YjdM, among the 390 proteins with four-digit EC numbers predicted by DeepECtransformer. DeepECtransformer showed improved performance compared to previous methods, such as sequence alignment, not only in predicting the correct EC numbers for mis-annotated enzymes but also in predicting EC numbers for proteins with poor characterization. Moreover, DeepECtransformer was employed to re-evaluate the EC numbers of 128,100,490 protein sequences in 70,600 genomes in the NCBI genome database (Supplementary Data 6), introducing the potential for characterizing previously unknown enzyme functions. For example, this resource has the potential to bridge the knowledge gap between biochemical reactions and their associated genes by identifying the enzymes responsible for catalyzing these reactions. In the BRENDA database (v. 1.1.0), 2293 out of a total of 7753 EC numbers lack annotated protein information[42]. DeepECtransformer generated 7,694,772 putative protein sequences across 47,692 genomes for 271 of these EC numbers. These results provide a valuable resource for discovering unknown metabolic functions by enriching the pool of sequences available for future analysis.

In addition to predicting protein characteristics from a primary sequence, it is also crucial to understand the sequence features that impact enzyme function. Recent studies on biological sequences using deep learning have attempted to interpret the inner workings of neural networks through various approaches[17–22]. In this study, we conducted an analysis of the self-attention layers within the DeepECtransformer neural network to identify the specific features that AI focuses on to classify enzyme functions. Our results showed that the AI effectively detects functional regions such as active sites and ligand interaction sites, in addition to well-known functional domains such as Pfam domains. These identified motifs have the potential to enhance our understanding of enzyme functions. Our analysis successfully visualized commonly highlighted motifs consisting of 16 amino acid residues, but it is possible that DeepECtransformer may also be capable of detecting longer-range protein interactions. As an example, the self-attention layer of DeepECtransformer identified multiple ligand interaction residues spanning the entire sequence of *E. coli*'s malate dehydrogenase (Supplementary Fig. 8). Further application of alternative interpretation methods holds the promise of uncovering previously unknown yet critical features of enzymes[17–22].

The use of EC numbers as the standard classification scheme for enzyme functionality has been well-established and continually updated. However, the four-digit structure of EC numbers often introduces ambiguity and makes it challenging to provide clear descriptions of metabolic functions using data-driven classifiers. For instance, a single metabolic reaction can be described by multiple EC numbers (e.g., malate dehydrogenase can be EC:1.1.1.37 and EC:1.1.1.375), and some EC numbers may represent general types of enzymes with ambiguous metabolic reactions (e.g., alcohol dehydrogenase, EC:1.1.1.1). Furthermore, the EC classification scheme is constrained by known substrates and reactions, posing difficulties in extrapolating classifiers for enzymes whose substrates are not clearly known. While GO terms are widely used for protein functionality classification, they may not comprehensively cover all metabolic reactions. As of July 2023, out of the 8056 EC numbers with complete four digits, only 5216 had corresponding GO terms (http://current.geneontology.org/ontology/external2go/ec2go). Hence, establishing a clear scheme for describing metabolic reactions could significantly enhance our understanding of enzyme functionality.

In conclusion, we expect DeepECtransformer, as a tool for predicting EC numbers, to become widely used in functional genomics. Its capabilities allow for the analysis of metabolism at a systems level, facilitating the construction of comprehensive genome-scale metabolic models, potentially minimizing gaps or missing information.

## Methods

### Reagents

All oligonucleotides were purchased from Genotech (Daejeon, Korea), and gene sequencing was performed at Macrogen (Daejeon, Korea). All enzymes and reagents for DNA manipulation were purchased from New England Biolabs (Berverly, MA, USA), TaKaRa Shuzo (Shiga, Japan), and Sigma-Aldrich (St. Louis, MO, USA). DNA fragments and plasmid DNA were purified using Qiagen kit (Qiagen, Chatsworth, CA, USA). All chemicals used in this study were purchased from either Sigma-Aldrich or Junsei Chemical (Tokyo, Japan).

### Construction of plasmids and strains

The strains, plasmids, and oligonucleotides used in this study are listed in Supplementary Table 5. To construct pET22b(+)-ygfF, pET22b(+)-yciO, and pET22b(+)-yjdM, pET22b(+) was linearized using *Ava*I and *Nde*I. *ygfF*, *yciO*, and *yjdM* gene fragments were prepared by polymerase chain reaction using the primers P3/P4, P5/6, and P7/8, respectively, using genomic DNA of *E. coli* MG1655 as a template. Prepared gene fragments were then ligated with linearized pET22b(+) using Gibson assembly. Correct vector construction was verified using DNA sequencing. Plasmids pET22b(+)-ygfF, pET22b(+)-yciO, and pET22b(+)-yjdM were then transformed to *E. coli* BL21 (DE3) strain resulting in BL21 (DE3) (pET22b(+)-ygfF), BL21 (DE3) (pET22b(+)-yciO), and BL21 (DE3) (pET22b(+)-yjdM) strains.

### Purification of YgfF, YciO, and YjdM

*E. coli* BL21 (DE3) strains harboring pET22b(+)-ygfF, pET22b(+)-yciO, and pET22b(+)-yjdM were cultured for the overexpression of C-terminus his-tagged YgfF, YciO, and YjdM in 500 ml of LB medium at 37 °C. The expression of his-tagged YgfF, YciO, and YjdM were induced by adding 1 mM IPTG after 3 h of cultivation. Cells were collected after additional 6 h of cultivation by centrifugation at $2090 \times g$ for 15 min. Harvested cells were suspended in 30 ml of equilibrium buffer that consists of 50 mM NaH2PO4, 0.3 M NaC1, 10 mM imidazole (pH 7.5). The cells were disrupted using ultrasonic homogenizer (VCX-600; Sonics and Materials Inc., Newtown, CT) with a titanium probe 40 T (Sonics and Materials Inc.). Cell debris was separated by centrifugation at $15,044 \times g$ for 40 min, and the resulting supernatants were loaded onto Talon metal affinity resin (Clontech, Mountain View, CA). Equilibrium buffer supplemented with 10 mM of imidazole (5 ml) was flown through the resin to elute his-tagged YgfF, YciO, and YjdM. Finally, the buffer solution of the eluted protein was changed to their reaction buffers by using Amicon Ultra-15 Centricon (Millipore, Beilerica, MA) with a pore size of 10 kDa. The composition of each reaction buffer is: glucose dehydrogenase assay buffer (Sigma-Aldrich, St. Louis, MO, USA) for YgfF, a mixture of 50 mM MOPS, 20 mM MgCl$_2$, 25 mM KCl, and 20 mM NaHCO$_3$ for YciO and 50 mM Tris-HCl (pH 8.0) for YjdM. The concentrations of the purified YgfF, YciO, and YjdM were measured by the Bio-Rad Protein Assay Kit (Bio-Rad, Hercules, CA) using bovine serum albumin (BSA) as a standard.

## In vitro enzyme assay

The reaction mixture for YgfF is composed of 82 μl of glucose dehydrogenase (GDH) assay buffer, 8 μl of GDH developer, and 10 μl of 2 M glucose were used, and supplemented with 50 μl of the purified his-tagged YgfF. The enzyme reaction was carried out for 3 min at 37 °C and measured $A_0$. After 30 min, $A_1$ was measured using a spectrophotometer at $OD_{450}$. Enzyme activity was measured total GDH amount using a GDH colorimetric kit (Cat# K786-100; BioVision, Milpitas, CA). For absolute quantification of the GDH concentration, a standard curve was prepared according to the manufacturer's protocol. The reaction mixture for YciO is composed of 2 μl of 200 mM ATP, 2 μl of 1 M L-threonine, 8 μl of the purified his-tagged YciO, and 188 μl of reaction buffer mentioned above. The enzyme reaction was carried out for 20 min at 25 °C. Enzyme activity was determined by measuring the total inorganic pyrophosphate ($PP_i$) level using a $PP_i$ assay kit (Cat# MAK386, Sigma-Aldrich). Each well of the 96-well plate was read using a spectrophotometer at $OD_{570}$. For absolute quantification of the $PP_i$ concentration, a standard curve was prepared according to the manufacturer's protocol. The reaction mixture for YjdM is composed of 94 μl of 50 mM Tris-HCl (pH 8.0), 2 μl of 10 mM phosphonoacetic acid, and 4 μl of the purified his-tagged YjdM. The enzyme reaction was carried out for 30 min at 35 °C. Enzyme activity was determined by measuring the total phosphate level using a phosphate assay kit (Cat# MAK308, Sigma-Aldrich). Each well of the 96-well plate was read using a spectrophotometer at $OD_{620}$. For absolute quantification of the phosphate concentration, a standard curve was prepared according to the manufacturer's protocol. All the in vitro enzyme assays were conducted in triplicates.

## Dataset construction

The amino acid sequences of enzymes were retrieved from UniProt Knowledgebase (UniProtKB)/Swiss-Prot and TrEMBL entries[27] which were released in April 2018. Sequences are processed to filter out (i) sequences without all four EC digits, (ii) sequences with non-standard amino acids, (iii) sequences that are longer than 1000 amino acids (which only account for 3.56% of the dataset), (iv) redundant sequences, and (v) sequences of which EC number has less than 100 sequences in the dataset. The processed dataset, called uniprot dataset, contains the amino acid sequences of 22,477,695 enzymes which cover 2802 EC numbers. The dataset was randomly split into a training dataset, validation dataset, and test dataset by the ratio of 8:1:1. All of the split datasets contain the whole EC number types (2802 EC numbers). For the homologous enzyme search, we used the amino acid sequences of enzymes in Swiss-Prot entries which contain at least one EC number. The processed dataset, called the swissprot dataset, contains the amino acid sequences of 226,325 enzymes which cover 5179 EC numbers including EC numbers without all four EC digits. To compare the performance of EC number prediction tools, we used NEW-392 dataset, which contains 392 amino acid sequences covering 177 types of EC numbers, and Price-149 dataset, which contains 149 amino acid sequences covering 56 types of EC numbers, provided by Yu et al.[24].

## Neural network architecture and training process

DeepECtransformer employs a neural network to predict the EC numbers of enzymes. The neural network uses the BERT architecture which encodes the input amino acid sequences of enzymes using self-attention layers (Fig. 1a)[26,43]. In this work, we modified the pretrained ProtBert model pretrained on UniRef100[8]. The neural network takes a protein sequence which is tokenized by each amino acid. The tokens are embedded into vectors with 128 dimensions, which are fed into two subsequent self-attention encoders. Each self-attention encoder contains a multi-head self-attention layer with eight heads. The number of hidden nodes of feed-forward layers in a multi-head self-attention layer is 128, and the number of hidden nodes of feed-

forward layers in a self-attention intermediate layer is 256. The output of the BERT is processed by two convolutional layers with 128 filters of which sizes are (4, 128) and (4, 1), respectively for each layer. Batch normalization and rectified linear unit (ReLU) activation layers were used after each convolutional layer. At the end of the network, a max-pooling layer, a linear layer and a sigmoid layer were used to classify the corresponding EC numbers of the embedded representation. The neural network was trained on the training dataset for 30 epochs using an Adam optimizer with a learning rate of 0.001, decaying with a factor of 0.95 for every epoch. The batch size was 512. Focal loss with a tunable focusing parameter ($\gamma$) 1.0 was used as a loss function[36]. For the inference, an EC number is assigned to the input sequence if the output score, calculated from the sigmoid layer, exceeds a threshold. In the case of promiscuous enzymes, all EC numbers exceeding their respective thresholds were assigned as the predicted EC numbers. Using the validation dataset, the optimal threshold for each EC number was searched that maximized the $F_1$ score of the EC number. The evaluation metrics are calculated as follows.

$$Macro\ precision = \frac{1}{C}\sum_{i=1}^{C}\frac{TP_i}{TP_i+FP_i},$$

$$macro\ recall = \frac{1}{C}\sum_{i=1}^{C}\frac{TP_i}{TP_i+FN_i}, \quad macro\ F_1\ score = \frac{1}{C}\sum_{i=1}^{C}\frac{2\bullet Precision_i\bullet Recall_i}{Precision_i+Recall_i},$$

$$micro\ precision = \frac{\sum_{i=1}^{C}TP_i}{\sum_{i=1}^{C}(TP_i+FP_i)}, \quad micro\ recall = \frac{\sum_{i=1}^{C}TP_i}{\sum_{i=1}^{C}(TP_i+FN_i)},$$

$$micro\ F_1\ score = \frac{2\bullet micro\ precision\bullet micro\ recall}{micro\ precision+micro\ recall},$$ where $TP_i$ is the number of true positive prediction for class $i$, $FP_i$ is the number of false positive prediction for class $i$, $FN_i$ is the number of false negative prediction for class $i$, and $C$ is the number of classes. Source code for DeepECtransformer is available at https://github.com/kaistsystemsbiology/DeepProZyme.

## Homologous enzyme searches

We used DIAMOND v2.0.11 to search for homologous enzyme[28]. The minimum percent of sequence identity and coverage were optimized by searching 361 parameter sets (sequence identity and coverage in 5% steps from 5% to 95%) (Supplementary Fig. 9)[4]. The swissprot dataset was randomly divided into queries (113,162 sequences) and a reference database (113,163 sequences) for the optimal parameter set searching. Using the parameter sets, we aligned the queries on the reference database to assign queries with homologous enzymes and the corresponding EC numbers. A parameter set with a minimum sequence identity of 50% and a minimum sequence coverage of 75% showed the highest micro $F_1$ score, 0.8739, was selected. The optimal parameter set was used for the DeepEC homologous search using the whole swissprot dataset as the reference database.

## Visualizing the latent space of the neural network

Latent representations of the amino acid sequences of 217,123 enzymes in the Swiss-Prot database were visualized using the TMAP algorithm[30] and Faerun library[44,45]. The embedded latent vectors before the last linear layer were used as the latent representations. To construct a tree map that compares the Swiss-Prot annotations and DeepECtransformer predictions, 179,655 enzyme entries that have at least one predicted EC number by the neural network were used.

## Analysis of attention layers

Latent representations of the amino acid sequence of an enzyme after self-attention layers were calculated using the DeepECtransformer neural network. Each attention head learns different inter-residue dependencies using self-attention[26]. The highlighted residues of each attention head were analyzed using attention scores, which are the average attention values of each residue in the attention map. The attention scores were visualized into sequence logos

using Logomaker[46]. Because the former self-attention layer only captured inter-residue dependencies between proximal residues, the averaged values of residues in the amino acid sequence of an enzyme did not show any specifically highlighted residues. Therefore, the highlighted residues are analyzed using the latter self-attention layer in this study.

Common motifs used for each EC number were extracted using the amino acid sequences of enzymes in Swiss-Prot entries having sequence lengths of 50–1000 without any non-canonical amino acid. The amino acid sequences of enzymes which have EC numbers that cannot be predicted by DeepECtransformer were also excluded. For each attention head in the second self-attention layer, a motif having the length of 16 amino acid residues was extracted, of which a residue having the maximum attention score is centered. Cluster analysis was performed for motifs of each EC number using MMseqs2[47]. To get the representative common motifs for the clusters, multiple sequence alignment was performed for the top 5 clusters having the largest motifs per EC number[48]. The identified representative common motifs are visualized using Logomaker[46].

### Processing 323,985 protein sequences of 1122 *E. coli* strains
The pan-genome of 1122 publicly available strains of *E. coli* was constructed by clustering protein sequences based on their sequence homology using the CD-hit package (v4.6)[49]. CD-hit clustering was performed with 0.8 threshold for sequence identity and a word length of 5. Clusters by CD-hit were used as representative gene families and the associated strain-specific sequences per gene family comprise the allele set studied.

### Phylogeny analysis of protein variants
The phylogeny of *aroL*, *lsrF*, and *ttdA* variants was analyzed using ClustalW[48] and the phylogenetic trees were visualized using iTOL[50].

### Program environment
All the development and analysis of DeepECtransformer were implemented using Python 3.6 under Ubuntu 16.04 LTS. The neural network of DeepECtransformer was trained and executed on NVDIA Tesla V100 GPUs. The following Python modules were used in this study: biopython v1.78, numpy v1.17.3, pandas v0.25.2, CD-hit package v4.6, Cluster Omega v1.2.3, DIAMOND v.2.0.11, faerun v0.3.20, logomaker v0.8, matplotlib v3.2.2, MMseq2, pytorch v1.7.0, scikit-learn v0.21.3, tmap v1.0.4, and transformers v3.5.1.

### Reporting summary
Further information on research design is available in the Nature Portfolio Reporting Summary linked to this article.

## Data availability
Supplementary Data 1–7 are available at https://doi.org/10.5281/zenodo.10023678 (ref. 51). Source data are provided with this paper and also available from Figshare: https://doi.org/10.6084/m9.figshare.23577036 (ref. 52). UniProtKB/Swiss-Prot and TrEMBL entries (released in April 2018) were used to construct the uniprot dataset. Protein 3D structures used in this paper can be accessed with PDB ID 1IB6 and 1CIV. Source data are provided with this paper.

## Code availability
The program is available at https://github.com/kaistsystemsbiology/DeepProZyme (ref. 53).

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

## Acknowledgements

This work is supported by the Development of next-generation biorefinery platform technologies for leading bio-based chemicals industry project (2022M3J5A1056072) and by Development of platform technologies of microbial cell factories for the next-generation biorefineries project (2022M3J5A1056117) from National Research Foundation supported by the Korean Ministry of Science and ICT.

## Author contributions

S.Y.L. conceived and designed the experiments. J.Y.K. and J.A.L. performed the experiments. G.B.K., C.J.N., B.O.P. and S.Y.L. analyzed the data. G.B.K. contributed analysis tools. G.B.K., J.Y.K., J.A.L., C.J.N., B.O.P., and S.Y.L. wrote the paper.

## Competing interests

The authors declare no competing interests.
