## [Peer Review File · Nature Communications]

Functional annotation of enzyme-encoding genes using deep learning with transformer layersReviewer #1 (Remarks to the Author):

The study reported in the manuscript developed a Machine Learning (ML) model, called DeepECtransformer for prediction of Enzyme Commission (EC) numbers of proteins from their amino-acid sequences. The authors use the developed method to predict the EC numbers of previously un-annotated genes from the Escherichia coli K-12 MG1655 genome. They further characterize in vitro enzymatic activities of three selected genes and find that the activities match EC numbers predicted by DeepECtransformer. The authors also comment upon the interpretability of the trained models by analyzing attention weights of the transformer. Overall, the study presented could be a potentially useful one to the field, but the evaluation procedures are hard to assess making it difficult to assess its true effectiveness. Also, no specific demerits of multiple existing tools that DeepECtransformer aims to replace are described. Furthermore, some of the claims made in the manuscript are not well supported. The authors are asked to consider the following points:

Concerns/Suggestions:

1. The biggest concern with this study is the lack of a complete description of the evaluation procedures. The authors note that they randomly split the uniprot dataset into train, validation, and test splits (Lines 413-414). One would expect thus that this would result in a single test dataset and a set of evaluation metrics for the same viz., one number each for Precision, Recall and F1-score. However, the authors show boxplots of metrics for evaluation on the test set (Fig. 1b). And there is no clear explanation anywhere as to what the metrics in these boxplots represent. Do the authors perform any cross-validation? Or do they repeat the evaluation procedure multiple times with different trained models to account for the stochasticity of training? Moreover, the inter-quartile ranges for the boxplots are unusually large leaving the impression that the prediction performances are highly inconsistent. Without systematic and transparent evaluation procedures, any claims on the successes of Machine Learning methods could be misleading and can lead to spurious conclusions. The authors should consult existing literature (such as (Ref 1: P Radivojac et. al.) to employ up to date methods for evaluating protein-function prediction.
2. The authors state - "DeepECtransformer demonstrated an improved ability to predict EC numbers for enzymes that have low sequence identities to those in UniProtKB/Swiss-Prot database" (Lines 101-103). Why do the authors measure sequence identities against only the UniProtKB/Swiss-Prot when they use a training dataset which also contains TrEMBL entries (Line 408 and Extended Data Fig. 2)? The authors should consider sequence similarities against the entire training dataset. The key point of such evaluation is to see if the model would perform well on sequences dissimilar from those used in the training dataset. The way it is done currently is incomplete and can result in misleading conclusions. For example, the authors make a statement that their method has the capability to predict enzyme functions for proteins with low sequence identities to known enzymes (Lines 200-201). Again, 'known enzymes' in this statement should refer to 'enzymes seen by the model as part of training (i.e., entire training dataset)' not just UniProtKB/Swiss-Prot. The authors can refer to existing literature such as T Sanderson et. al. (Ref 2). The authors use the conclusions drawn from this evaluation procedure to assume that their method can be used to uncover functions of enzymes that have not been characterized till date which is not justified (Lines 200-212).
3. The authors state that DeepECtransformer predicts all four digits of EC numbers for 295 proteins of E. coli K-12 MG1655. From these, they experimentally characterize the in vitro activities of YgfF, YciO and YjdM to validate the predictions. While backing up computational predictions by experimental validation is highly desirable, more information is needed to justify the choice of these three genes out of the 295. Were any proteins other than these three tested which do not have experimental activities in line with the predictions? Is there some pattern that the authors find in correct and incorrect predictions made by DeepECtransformer?
4. In lines 96-97, the authors used a test dataset of 2,013,612 enzymes to perform comparative evaluation with other methods. For this evaluation, are the models trained separately by excluding these enzymes from the training dataset? The authors should provide more details on how this

test dataset was curated and on its overlap with the original test dataset (used for Fig. 1b) and/or training datasets. These details are crucial for understanding the predictive abilities of ML methods and hence cannot be ignored.

5. The authors note that DeepECtransformer has two engines: a neural network and a homology search against UniprotKB/Swiss-Prot. The authors use the homology-based prediction when the neural network fails to predict EC numbers (Line 80). What is the definition of 'neural network fails to predict'? This requires further explanation. Further, all the three genes YgfF, YciO and YjdM have homologous proteins in UniprotKB with $\geq 80\%$ sequence identities (Uniprot-ids: A0A377Z437, A0A663AY21 and A0A1E3L8Q4) that have the same predictions as those made by DeepECtransformer. This leaves the reviewer with the question of whether these predictions are made by DeepECtransformer using the neural network or by simply a homology search. This should be mentioned in the manuscript. Also, are these homologous proteins part of the dataset used for training the prediction model? This detail should be included, and this is required for any test predictions talked about in the manuscript.

6. In Lines 213-214, the authors write - "To validate the ability of DeepECtransformer in identifying metabolic functions that cannot be detected by other methods ... ". What are these other methods? This statement is too vague and can be misleading. The authors should state what the other methods are explicitly. If appropriate, the authors should include why other methods cannot detect functions in these cases. Do other methods give no prediction at all in these cases due to some systematic shortcoming?

7. In Lines 104-105, the authors write - "The accuracy of DeepECtransformer was further demonstrated by its ability to correct mis-annotated EC numbers in UniProtKB". They illustrate this by quoting several hand-picked examples such as P93052, Q8U4R3 etc. As stated in bullet-point (4) above, the authors should note for each of these predictions if the sequence and the respective misannotated labels were a part of the training dataset or not. In this section (Lines 104-129), the authors only discuss only positive examples i.e., predictions made correctly by DeepECtransformer without any plausible explanations for the successful predictions. Also, the differences in mis-annotations and correct-annotations are not large (i.e., 1.1.1.27 vs. 1.1.1.37 and 1.3.1.14 vs. 1.3.98.1). The authors can provide some insight into the success of such predictions using the attention weights. Also, the authors could check if the misannotated sequences have very similar sequences in training that are correctly annotated. This would explain why DeepECtransformer is able to arrive at the right annotations. Following this, the authors state - "DeepECtransformer made predictions for 26,140 proteins with EC numbers that differed from those in Swiss-Prot ... Until such experimental validation results are accessible, the predictions made by DeepECtransformer can serve as the most plausible functions of these proteins.". All of this leaves the impression that DeepECtransformer's predictions are better than SwissProt's labels. The authors should note that even though a few annotations in SwissProt could be erroneous, they are the best available 'ground truth' labels which indeed comprise a good fraction of the training data here. This entire section can be misleading and should be changed appropriately.

8. The introduction section has no mention of existing EC prediction methods other than DeepEC. The authors need to highlight some existing methods along with their limitations (if any) to contrast with DeepECtransformer and/or any challenges that the current work specifically addresses. The authors should at least talk about some of the recent ones such as Ref. 2 and Ref. 3. Moreover, the authors themselves state in the manuscript (Lines 250-251), a recently developed method CLEAN showed improved performance compared to DeepECtransformer. Also, ProteInfer addresses several aspects that are not addressed in this study like no arbitrary sequence length cutoffs and the ability to predict more than just the EC number using GO labels. All this should be appropriately discussed.

9. Line 82 needs further explanation. How is the increase in coverage from 2,802 to 5,360 arrived at?

10. Extended Data Fig. 1 lacks clear description. How do the authors arrive at a distribution of number of sequences for each EC class (these should be single numbers?) ?

References

1. Radivojac, P., Clark, W., Oron, T. et al. A large-scale evaluation of computational protein function prediction. *Nat Methods* 10, 221–227 (2013)
2. Sanderson, T., Bileschi, M. L., Belanger and D., Colwell, L. J., ProteInfer, deep neural networks for protein functional inference. *eLife* 12:e80942 (2023)
3. Yu, T., Cui, H., Li, J. C., et al., Enzyme function prediction using contrastive learning. *Science* 379,1358-1363 (2023)

Reviewer #2 (Remarks to the Author):

In the work, Kim et al. developed a deep learning model to predict the EC number for enzymes, which were then applied to predict the function for the unannotated genes in *E. coli*. Altogether, the model can then be used to identify mis-annotated enzymes in database, identify functional regions in the enzyme, and also identify emerged functions during the evolution, making it a tool with broad applications in the biology field. Overall, the manuscript is well structured and organized, addressing relevant scientific questions enzyme function prediction. However, I have a few comments that might be helpful to improve the study.

1. The comparison between DeepECtransformer and the recently published CLEAN (*Science* 379, 1358–1363, 2023) doesn't appear to be sufficient. If the author claim that DeepECtransformer's capability to predict EC numbers for unannotated genes surpasses that of CLEAN, it is necessary to compare how many proteins with EC numbers can be predicted by CLEAN, in terms of the y-tome (1600 genes) mentioned by the author.
2. Additionally, the author's comparison of DeepECtransformer with other methods is limited to the Price-192 data, omitting the use of NEW-392 and the newly generated in-house data from CLEAN for comparison. Although the author mentioned that NEW-392 data was employed for HDMLF training, it remains essential to present a comparison between DeepECtransformer and CLEAN, as well as the second-best model ProteInfer (as indicated in the CLEAN paper in terms of this dataset). Furthermore, the second-best model BlastP, with precision of 0.5083, recall of 0.375, and an F1 score of 0.385 (as outlined in the CLEAN paper), also appears to outperform DeepECtransformer in terms of Price-192 data. This aspect should also be acknowledged, compared, or elucidated.
3. As mentioned in Line 421, there are some EC number not covered by DeepECtransformer, please list the reason and also compare the EC number coverage with CLEAN.
4. Line 214-215: an explanation for the rationale behind selecting only three genes for experimental validation is notably absent. Clarification regarding the basis for this particular choice would enhance the understanding of the experimental design.
5. It would be valuable to provide a justification for the chosen data split method. The current approach involves randomly dividing the dataset into training, validation, and test sets at an 8:1:1 ratio. However, considering the presence of numerous highly similar sequences within the UniProt database, there is a potential concern that this random split might inadvertently allocate highly similar sequences to both the training and test sets. If the objective is to develop a predictive model capable of generalizing effectively to data points that exhibit dissimilarity from the training data, it could be advisable to consider adopting a data split methodology akin to that employed in the CLEAN study. Alternatively, the author might consider evaluating sequence similarities between the training and test datasets.
6. One would expect that the EC class with more datapoints should achieve better performance in terms of prediction. Could the authors discuss this with results shown in Figure 2?
7. How does the authors handle enzymes with multiple EC numbers during the training and prediction?
8. May the author also discuss the limitation of this method?

Minor:

1. Line 420: Please add explanation for "NEW-392 dataset" and "Price-149 dataset".

2. In the extended Data Fig.2, the comparison with CLEAN should also be added.
3. Line 208: Please explain the reason for 74 proteins (464 protein – 390 proteins) with predicted EC numbers but does not contain four full digits.
4. Line 220, YciO was previously be annotated as the Suc5 family protein by other algorithms as shown in UniProt database. If so, please mention this in the text.

Reviewer #3 (Remarks to the Author):

This article introduces a neural network model, DeepECtransformer, to predict EC numbers of enzymes based on the amino acid sequences. The technique looks promising and will help the research community predict metabolic functions of organisms using just sequence information.

Overall, the extent of advance over their prior approach, deepEC, seems limited. The rationale and advantage of using transformer layers is never discussed. Limitations of this new strategy are also not discussed. What were the sources of errors with the original deepEC and in what cases does this new approach improve upon it? How exactly were the limitations of deepEC resolved with this new approach?

Also a new approach, called CLEAN, appears to outperform in every category. Why was CLEAN not able to make predictions for the two γ -genes? Were there cases where CLEAN made predictions and their approach didn't? This field is still developing, and new tools are needed. CLEAN and DeepEC are still work in progress, but it would help to clarify strengths/weaknesses of these methods. CLEAN should be reported in the main text and figures too.

How are enzymes with multiple activities (ECs) handled in training/testing and novel predictions?

There's a cryptic statement in the results, which is not described in detail, but seems very important in influencing its accuracy -

"If the neural network fails to predict EC numbers, homologous enzymes are analyzed and EC numbers are assigned using UniProtKB/Swiss-Prot enzymes as the subject database".

For how many cases did it fail? What scenarios does it commonly fail? What's the accuracy without the homology approach? Are failures considered as incorrect predictions?

Choice of test set: the authors randomly sampled 10% of the entire data. Issues of data leakage are not addressed when selecting an appropriate test set. What is the accuracy when there isn't a close homolog in the training? What's the accuracy in annotating a 'new' organism or a 'new' enzyme class?

How were 'misannotated' enzymes chosen for experimental testing? Currently, they seem to have arbitrarily picked 3 enzymes.

Minor comments

Line 109, they identified correct activities for some 'misannotated' enzymes; while they have shown proof for the new activity, they didn't show evidence that the original activity was incorrect. Maybe they are multifunctional/promiscuous enzymes? How was the original activity determined?

Line 179-197, related to E.coli strains, is there any evidence supporting these predictions?

Line 67: Authors mentioned that DeepECtransformer predicted 464 proteins out of 1569, why is that so? Why only 464? Is there a case where this model will fail in predicting the EC number (doesn't the homology approach handle that (line 80-81))?

Line 88: Inconsistent caption of the figure (1d) and description in text

Lines 110-114: Authors talk about mis annotations of proteins, Q8U4R3 and Q038Z3, and claim that their approach has accurately predicted it. Do the authors have any proof of the validity and correctness of their claim? Is there a confidence score associated with these predictions?

Line 208/215: Authors mention 390 proteins have full EC numbers, while later they mention 295 proteins have full EC numbers.

Lines 330-365 in methods, I'm unsure why it's there, this organism and bioreactor setup was not discussed in results.

Table 1 Define micro and macroprecision/recall/f1 in the Methods section.

Figure 1 – include deepEC architecture for context.

Fig 4 – include the specific activity tested (i.e. substrate/product)

Ex Fig 1 – what is the y axis? Needs a more detailed caption.

Ex Fig 9 – where is this analysis used? A longer caption is needed.

Responses to the Reviewers' comments

Reviewer #1 (Remarks to the Author):

The study reported in the manuscript developed a Machine Learning (ML) model, called DeepECtransformer for prediction of Enzyme Commission (EC) numbers of proteins from their amino-acid sequences. The authors use the developed method to predict the EC numbers of previously un-annotated genes from the Escherichia coli K-12 MG1655 genome. They further characterize in vitro enzymatic activities of three selected genes and find that the activities match EC numbers predicted by DeepECtransformer. The authors also comment upon the interpretability of the trained models by analyzing attention weights of the transformer. Overall, the study presented could be a potentially useful one to the field, but the evaluation procedures are hard to assess making it difficult to assess its true effectiveness. Also, no specific demerits of multiple existing tools that DeepECtransformer aims to replace are described. Furthermore, some of the claims made in the manuscript are not well supported. The authors are asked to consider the following points:

[Response] We appreciate your time and effort in reviewing our manuscript and providing invaluable comments to improve our manuscript. We have addressed your comments as follows.

Concerns/Suggestions:

1. The biggest concern with this study is the lack of a complete description of the evaluation procedures. The authors note that they randomly split the uniprot dataset into train, validation, and test splits (Lines 413-414). One would expect thus that this would result in a single test dataset and a set of evaluation metrics for the same viz., one number each for Precision, Recall and F1-score. However, the authors show boxplots of metrics for evaluation on the test set (Fig. 1b). And there is no clear explanation anywhere as to what the metrics in these boxplots represent. Do the authors perform any cross-validation? Or do they repeat the evaluation procedure multiple times with different trained models to account for the stochasticity of training? Moreover, the inter-quartile ranges for the boxplots are unusually large leaving the impression that the prediction performances are highly inconsistent. Without systematic and transparent evaluation procedures, any claims on the successes of Machine Learning methods could be misleading and can lead to spurious conclusions. The authors should consult existing literature (such as (Ref 1: P Radivojac et. al.) to employ up to date methods for evaluating protein-function prediction.

[Response] Thank you for the great comments. As the reviewer mentioned, we conducted a random split of the dataset into train, validation, and test datasets, and the evaluation was performed once without employing cross-validation. In Fig. 1b, the boxplots illustrate the performance of the neural network for individual EC number classes in the test dataset. Within each boxplot, each data point corresponds to the neural network's performance of a single EC number (e.g., EC:1.1.1.1), while each box (e.g., the box for EC:1) represents the collective performance across all EC numbers in that class (e.g., EC:1.-.-). According to your comment,

we added the following sentences to the Fig. 1 legend to clearly explain the boxplots: “The boxplots illustrate the varying performance of the neural network by the first-level EC numbers. Each data point in the boxplot represents the performance of the neural network for a single EC number.”. In the section, we aimed to analyze the varying performance of the neural network among different EC number classes. Due to imbalances in the number of sequences per EC number, some boxplots exhibit wide interquartile ranges. Regarding the reviewer’s comment, we have discussed the issue by exploring the relationship between the number of sequences per EC number and the network’s performance in lines 96-108 of the revised manuscript: “The performance results varied depending on the EC number class, with precision ranging from 0.7589 to 0.9506, recall ranging from 0.6830 to 0.9445, and F₁ score ranging from 0.6990 to 0.9469 (Fig. 1b). The evaluation metrics (precision, recall, and F₁ score) of the neural network were lowest for the EC:1 class (oxidoreductases), which made up 13.4% of enzymes in the uniprot dataset (Fig. 1c). Despite this, it had a higher number of individual fourth-level EC numbers, accounting for 25.7% of total EC numbers (Fig. 1d). The EC:1 class has a smaller number of sequences compared to other classes, leading to a dataset imbalance (one-way ANOVA test, p-value < 0.001) (Extended Data Fig. 1).” . We rewrote the sentences to “The performance results varied depending on the EC number class, with precision ranging from 0.7589 to 0.9506, recall ranging from 0.6830 to 0.9445, and F₁ score ranging from 0.6990 to 0.9469 (Fig. 1b). The evaluation metrics (precision, recall, and F₁ score) of the neural network were lowest for the EC:1 class (oxidoreductases), which made up 13.4% of enzymes (i.e., 313,328 sequences) and made up 25.7% of EC numbers (i.e., 720 EC numbers) in the uniprot dataset (Fig. 1c, d). The low performance for EC:1 class resulted from the inherent dataset imbalance, as the EC:1 class exhibited the lowest average number of sequences per EC number, with an average of 4,352 sequences compared to the other EC number classes (ranging from 6,819 sequences for EC:3 to 16,525 sequences for EC:6). Additionally, a statistical analysis of the data distribution confirmed that EC numbers belonging to the EC:1 class generally had fewer sequences compared to other EC number classes (one-way ANOVA test, p-value < 0.001) (Extended Data Fig. 1).” to clearly describe the analysis.

2. The authors state - “DeepECtransformer demonstrated an improved ability to predict EC numbers for enzymes that have low sequence identities to those in UniProtKB/Swiss-Prot database” (Lines 101-103). Why do the authors measure sequence identities against only the UniProtKB/Swiss-Prot when they use a training dataset which also contains TrEMBL entries (Line 408 and Extended Data Fig. 2)? The authors should consider sequence similarities against the entire training dataset. The key point of such evaluation is to see if the model would perform well on sequences dissimilar from those used in the training dataset. The way it is done currently is incomplete and can result in misleading conclusions. For example, the authors make a statement that their method has the capability to predict enzyme functions for proteins with low sequence identities to known enzymes (Lines 200-201). Again, ‘known enzymes’ in this statement should refer to ‘enzymes seen by the model as part of training (i.e., entire training

dataset)' not just UniProtKB/Swiss-Prot. The authors can refer to existing literature such as T Sanderson et. al. (Ref 2). The authors use the conclusions drawn from this evaluation procedure to assume that their method can be used to uncover functions of enzymes that have not been characterized till date which is not justified (Lines 200-212).

[Response] Thank you for the valuable comments. To address this concern, we calculated the sequence similarity between the entire training dataset and test dataset to re-evaluate the performance of EC number prediction tools. We confirmed that DeepECtransformer still shows an improved ability to predict EC numbers for enzymes with relatively low sequence similarities to those present in the training dataset. Accordingly, we replaced the Extended Data Fig. 3 and rewrote the sentence “Moreover, DeepECtransformer demonstrated an improved ability to predict EC numbers for enzymes that have low sequence identities to those in the UniProtKB/Swiss-Prot database.” to “Moreover, DeepECtransformer demonstrated an improved ability to predict EC numbers for enzymes that have low sequence identities to those in the training dataset.” in the lines 118-120 of the revised manuscript. Also, the sentence in the lines 219-220 was revised from “As discussed earlier, DeepECtransformer has shown the capability to predict enzyme functions for proteins with low sequence identities to known enzymes.” to “As discussed earlier, DeepECtransformer has shown the capability to predict enzyme functions for proteins with low sequence identities to enzymes seen by the model during the training.” according to the reviewer’s comment.

3. The authors state that DeepECtransformer predicts all four digits of EC numbers for 295 proteins of E. coli K-12 MG1655. From these, they experimentally characterize the *in vitro* activities of YgfF, YciO and YjdM to validate the predictions. While backing up computational predictions by experimental validation is highly desirable, more information is needed to justify the choice of these three genes out of the 295. Were any proteins other than these three tested which do not have experimental activities in line with the predictions? Is there some pattern that the authors find in correct and incorrect predictions made by DeepECtransformer?

[Response] Thank you for the comments. We aimed to show that DeepECtransformer can predict a wide range of EC numbers. To achieve this, we first selected three representative EC number classes for experimental validation: oxidoreductase (EC:1), transferase (EC:2), and hydrolase (EC:3). To validate the predicted enzyme function using *in vitro* enzyme assay, we identified 179 proteins from 254 proteins (comprising oxidoreductase, transferase, and hydrolase) that were predicted to be soluble by NetSolP [PMID: 35088833]. From the 179 proteins, we randomly selected three target proteins (i.e., YgfF, YciO, and YjdM) each from EC:1, EC:2, and EC:3. To elaborate on the processing step, we replaced the sentences “To validate the ability of DeepECtransformer in identifying metabolic functions that cannot be detected by other methods, we performed *in vitro* enzyme assays for three proteins, YgfF, YciO, and YjdM, among the 295 proteins that are exclusively predicted by DeepECtransformer to have 216 all four digits of EC numbers.” to “To validate the ability of DeepECtransformer in identifying metabolic functions that cannot be detected by DeepEC and Swiss-Prot functional annotation processes, we

performed *in vitro* enzyme assays to validate the predicted enzyme functions. We first selected three representative EC number classes, namely oxidoreductase (EC:1), transferase (EC:2), and hydrolase (EC:3) to show the capability of DeepECtransformer to predict unknown enzyme functions. Among the 295 proteins that are exclusively predicted by DeepECtransformer to have all four digits of EC numbers, 179 proteins are predicted to be soluble in *E. coli* by NetSolP, a deep learning model for protein solubility prediction (Supplementary Data 7)³¹. From the 179 proteins, we randomly selected three proteins, YgfF, YciO, and YdjM, that are predicted to be oxidoreductase, transferase, and hydrolase, respectively.” in lines 233-241 of the revised manuscript. Also, we included the prediction results of NetSolP for the 295 candidate proteins as Supplementary Data 7. Among them, we only tested three proteins, which were shown to have the predicted functions. We did not perform more experiments and thus did not find any that show incorrect predictions. However, it is possible that the DeepECtransformer may make incorrect predictions due to limited training dataset, which has been discussed in lines 96-111 in the revised manuscript.

4. In lines 96-97, the authors used a test dataset of 2,013,612 enzymes to perform comparative evaluation with other methods. For this evaluation, are the models trained separately by excluding these enzymes from the training dataset? The authors should provide more details on how this test dataset was curated and on its overlap with the original test dataset (used for Fig. 1b) and/or training datasets. These details are crucial for understanding the predictive abilities of ML methods and hence cannot be ignored.

[Response] Thank you. The test dataset, derived from the random split of the uniprot dataset, was curated. We only included protein sequences having EC numbers that can be predicted by DeepEC and DIAMOND for fair comparison. In this section, we aimed to show the improved prediction performance of DeepECtransformer in comparison to two baseline models: DeepEC and DIAMOND. It is worth noting that even if there is any possibility that sequences within the curated test dataset were used for training DeepEC or included in the sequence database of DIAMOND, DeepECtransformer showed better performance. According to your comment, sentences in lines 112-116 “The performance of DeepECtransformer was evaluated by comparing it with two other methods: DeepEC and a homology-based search tool, DIAMOND^{4, 25}. A test dataset consisting of the amino acid sequences of 2,013,612 enzymes, for which EC numbers can be predicted by all three tools, was employed for the comparison.” were revised to clearly describe how the test dataset was curated: “The performance of DeepECtransformer was evaluated by comparing it with two baseline methods: DeepEC and a homology-based search tool, DIAMOND^{4, 28}. For the comparison, the test dataset that has been used for the evaluation of the DeepECtransformer neural network was curated to consist of the amino acid sequences of 2,013,612 enzymes, for which EC numbers can be predicted by all three tools.”.

5. The authors note that DeepECtransformer has two engines: a neural network and a homology search against UniprotKB/Swiss-Prot. The authors use the homology-based prediction when the

neural network fails to predict EC numbers (Line 80). What is the definition of ‘neural network fails to predict’? This requires further explanation.

[Response] Thank you for the comments. We agree that the explanation of the use of homology search was insufficient. Following the reviewer’s comment, we replaced the sentence in lines 91-93 “If the neural network fails to predict EC numbers, homologous enzymes are analyzed and EC numbers re assigned using UniProtKB/Swiss-Prot enzymes as subject database^{4, 25}” to “If the neural network predicts no EC numbers for a given amino acid sequence, homologous enzymes for the amino acid sequence are analyzed using UniProtKB/Swiss-Prot enzymes as the subject database and EC numbers of the homologous enzymes are assigned^{4, 28}”.

Further, all the three genes YgfF, YciO and YjdM have homologous proteins in UniprotKB with $\geq 80\%$ sequence identities (Uniprot-ids: A0A377Z437, A0A663AY21 and A0A1E3L8Q4) that have the same predictions as those made by DeepECtransformer. This leaves the reviewer with the question of whether these predictions are made by DeepECtransformer using the neural network or by simply a homology search. This should be mentioned in the manuscript. Also, are these homologous proteins part of the dataset used for training the prediction model? This detail should be included, and this is required for any test predictions talked about in the manuscript.

[Response] Thank you for the important comments. For the enzymes YgfF, YciO, and YjdM, all predictions were made by the neural network with the prediction scores of 0.6331, 0.9108, and 0.6013, respectively. Additionally, we investigated whether homologous proteins of YgfF, YciO, and YjdM were present in the training dataset of the DeepECtransformer neural network. Despite the presence of homologous proteins for these three enzymes in the training dataset, it is important to note that the neural network’s predictions were not solely reliant on the sequence similarity between the query sequences and those in the training dataset. Specifically, the protein with the highest sequence identity to YgfF (EC:1.1.1.47) in the training dataset was A0A069CGU9_ECOLX, annotated with a different EC number (EC:1.1.1.100). Likewise, for YjdM, whose neural network-predicted EC number was EC:3.11.1.2 with a prediction score of 0.6103, the highest sequence identity match in the training dataset had a different EC number (C9Y1B8_CROTZ; EC:2.7.7.6). For YciO (EC:2.7.7.87), A0A2B7LMT3_9ESCH (EC:2.7.7.87) was the protein that was in the training dataset and has the highest sequence identity with YciO sequence. To further examine whether DeepECtransformer understands the functional region of YciO, the motifs assigned with high attention scores were analyzed. This analysis showed that the highlighted motifs corresponded to TIGR00057, a NCBIfam family for L-threonylcarbamoyladenylate synthase (Supplementary Fig. 4 of the revised manuscript). These findings show that DeepECtransformer leverages not only homology search but also incorporates latent features acquired during its training process. To provide these informations, we added the following sentences in lines 260-277 of the revised manuscript: “We also analyzed whether the EC number predictions for these three enzymes were made by the neural network or by the homology search. First, YgfF was predicted to have an EC number of EC:1.1.1.47 by the neural network with a prediction score of 0.6331. It should be noted that although YgfF exhibited a

higher sequence identity with a different enzyme (A0A069CGU9_ECOLX; EC:1.1.1.100) from the training dataset than glucose 1-dehydrogenase exhibiting the maximum sequence identity within the training dataset, the neural network achieved an accurate prediction. Likewise, for YjdM, predicted by the neural network as EC:3.11.1.2 with a prediction score of 0.6103, the training sequence with the highest sequence similarity had a different EC number (C9Y1B8_CROTZ; EC:2.7.7.6). Lastly, the predicted EC number for YciO by the neural network was EC:3.1.11.2 with a prediction score of 0.9108, and the training sequence with the highest sequence identity with YciO was also a L-threonylcarbamoyladenylate synthase (EC:3.1.11.2). To examine whether the neural network understands the functional regions of YciO rather than relying on the sequence identity, the motifs with high attention scores were analyzed. It was found that the highlighted motifs correspond to TIGR00057, a NCBIfam family for L-threonylcarbamoyladenylate synthase (Supplementary Fig. 4). These results suggest that DeepECtransformer leverages not only homology search but employs latent features learned during the training process during the prediction of EC numbers.”.

6. In Lines 213-214, the authors write - “To validate the ability of DeepECtransformer in identifying metabolic functions that cannot be detected by other methods ...”. What are these other methods? This statement is too vague and can be misleading. The authors should state what the other methods are explicitly. If appropriate, the authors should include why other methods cannot detect functions in these cases. Do other methods give no prediction at all in these cases due to some systematic shortcoming?

[Response] Thank you for the comments. We agree that the previous statement lacked sufficient explanation. Accordingly, we rewrote the sentence as follows: “To validate the ability of DeepECtransformer in identifying metabolic functions that cannot be detected by DeepEC and Swiss-Prot functional annotation processes, we performed *in vitro* enzyme assays to validate the predicted enzyme functions.”. We also added the following sentences in lines 277-281 to discuss why DeepEC and Swiss-Prot functional annotation processes were not able to predict the functions of some proteins: “Notably, DeepEC was unable to provide predictions for the three proteins, likely due to its low recall associated with predicting enzyme functions, stemming from a highly imbalanced dataset. Also, Swiss-Prot functional annotation failed to make predictions for the three proteins, possibly due to the limited sequence identity shared between the y-ome proteins and protein signatures available in existing databases.”. We also discussed the limitations of CLEAN in predicting enzyme functions for uncharacterized proteins which were caused by the use of contrastive learning in lines 303-311 of the revised manuscript: “However, while CLEAN showed improved performance compared to DeepECtransformer, it was not able to predict the EC numbers of YgfF and YjdM. Also, the use of CLEAN for the annotation of uncharacterized proteins requires careful interpretation of the prediction results because CLEAN assigns EC numbers for any input amino acid sequences, including non-enzyme amino acid sequences. CLEAN provides the confidence level (i.e., high, medium, low) of the predictions using a Gaussian mixture model. However, as the confidence level does not provide a detailed

interpretation of how AI performs the reasoning process, careful inspection of the predictions should be conducted for the analysis of uncharacterized proteins, especially when the uncharacterized proteins contain non-enzyme proteins.”.

7. In Lines 104-105, the authors write – “The accuracy of DeepECtransformer was further demonstrated by its ability to correct mis-annotated EC numbers in UniProtKB”. They illustrate this by quoting several hand-picked examples such as P93052, Q8U4R3 etc. As stated in bullet-point (4) above, the authors should note for each of these predictions if the sequence and the respective misannotated labels were a part of the training dataset or not.

In this section (Lines 104-129), the authors only discuss only positive examples i.e., predictions made correctly by DeepECtransformer without any plausible explanations for the successful predictions. Also, the differences in mis-annotations and correct-annotations are not large (i.e., 1.1.1.27 vs. 1.1.1.37 and 1.3.1.14 vs. 1.3.98.1). The authors can provide some insight into the success of such predictions using the attention weights. Also, the authors could check if the misannotated sequences have very similar sequences in training that are correctly annotated. This would explain why DeepECtransformer is able to arrive at the right annotations.

[Response] Thank you for the comments. As the reviewer commented, we added explanations on the attention weights of the example proteins and examined the sequence similarity between the example proteins and the training dataset. These details are provided in Supplementary Note 2. We have cited this supplementary information in the revised manuscript lines 140-143: “As the EC numbers suggested by DeepECtransformer are predictions, we further analyzed how AI made the predictions *in silico* (Supplementary Note 2). Even though *in silico* analysis of the predictions can provide clues for the predicted functionality, it is necessary to experimentally validate their functions.”.

Supplementary Note 2. Interpretation of DeepECtransformer predictions for the enzymes with mis-annotated EC numbers

DeepECtransformer predicted the EC numbers that are different from the annotations in UniProtKB/Swiss-Prot. To understand these predictions, we examined the attention scores computed in the self-attention layers. For instance, P93052, which was annotated as an L-lactate dehydrogenase (EC:1.1.1.27), was predicted as malate dehydrogenase (EC:1.1.1.37) by DeepECtransformer with the neural network prediction score of 0.9703. Upon further investigation, we identified a protein A0A147JD58_9SPHN, which was in the training dataset of the DeepECtransformer neural network, with the same EC number (EC:1.1.1.37) and a high sequence identity (96.6%). To confirm whether the prediction was solely made by the high sequence identity, motifs assigned with high attention scores were analyzed (Supplementary Fig. 5). We found that the highlighted motifs were shown in TIGR01763, an NCBIfam family associated with malate dehydrogenase. This demonstrates that DeepECtransformer not only relies on sequence similarity but also identifies motifs essential for the enzyme functionality. Similarly, we analyzed how DeepECtransformer predicted the enzyme function of Q038Z3 as

dihydroorotate dehydrogenase (NAD) (EC:1.3.1.14), which was previously annotated as dihydroorotate dehydrogenase (fumarate) (EC:1.3.98.1). DeepECtransformer neural network predicted Q038Z3 as dihydroorotate dehydrogenase (NAD) with the prediction score of 0.9791. Although dihydroorotate dehydrogenase (NAD) sequence that had the highest sequence identity was A0A508Z5W5_LACRH, of which sequence identity being 86.6%, the protein sequence in the training dataset that had the highest sequence identity was S2S6H3_LACPA (99.6%), which was annotated as dihydroorotate dehydrogenase (fumarate). When the motifs assigned with high attention scores were analyzed, conserved residues of cd04740, a CDD domain for DHOD_1B_like domain that represents the dihydroorotate dehydrogenase class 1B FMN-binding domain, were observed (Supplementary Fig. 6). We further analyzed the rationale of DeepECtransformer predictions for Q9WVK7 and C9K7D8. Even though the highest sequence identity with Q9WVK7 and NADP-dependent 3-hydroxybutyryl-CoA dehydrogenase sequences in the training dataset was 51.7% (A0A518EY01_9BACT), DeepECtransformer predicted EC numbers of Q9WVK7 as EC:1.1.1.157 (NADP-dependent 3-hydroxybutyryl-CoA dehydrogenase) with the prediction score of 0.6765, along with the annotated EC number (EC:1.1.1.35; 3-hydroxyacyl-CoA dehydrogenase). When the motifs with high attention scores were analyzed, a motif for NADP-binding proteins, [VILF]-X-G-X-[GSA]-X₂-[GAS]-X₆-[LAIFWCG], was observed (Supplementary Fig. 7)⁵. For C9K7D8, which has the highest sequence similarity of 49.4% (Q4WKQ2_ASPFU; EC:2.6.1.42) with the protein sequences in the training dataset, DeepECtransformer predicted the protein as a branched-chain amino acid transaminase (EC:2.6.1.42) with the prediction score of 0.9711. The motifs with high attention scores were shown in PTHR42825, a PANTHER family for amino acid aminotransferase (Supplementary Fig. 8). The EC number for Q8U4R3 was predicted as EC:4.4.1.15 (a D-cysteine desulhydrase) with a prediction score of 0.8477, which was originally annotated as 1-aminocyclopropane-1-carboxylate deaminase (EC:3.5.99.7). The protein sequence in the training dataset that has the highest sequence identity with Q8U4R3 was A0A162MS80_9FIRM (EC:4.4.1.15), having a sequence identity of 51.1%. Because the metabolic reaction of 1-aminocyclopropane-1-carboxylate deaminase is a sub-reaction of the metabolic reaction of D-cysteine desulhydrase, we were unable to identify motifs with high attention scores that specifically represent D-cysteine desulhydrase, rather than 1-aminocyclopropane-1-carboxylate deaminase.

Following this, the authors state – “DeepECtransformer made predictions for 26,140 proteins with EC numbers that differed from those in Swiss-Prot ... Until such experimental validation results are accessible, the predictions made by DeepECtransformer can serve as the most plausible functions of these proteins.”. All of this leaves the impression that DeepECtransformer’s predictions are better than SwissProt’s labels. The authors should note that even though a few annotations in SwissProt could be erroneous, they are the best available ‘ground truth’ labels which are indeed comprise a good fraction of the training data here. This entire section can be misleading and should be changed appropriately.

[Response] Thank you for the comments. We agree. According to your suggestion, we rewrote the section to tone down the content as follows: “The predictions made by DeepECtransformer can provide candidate entries for further review, contributing to the construction of a more robust knowledgebase.”. We also rewrote the sentence “Also, the improved performance of DeepECtransformer has enabled the correction of mis-annotated EC numbers in the UniProt Knowledgebase (UniProtKB).” to “Also, the improved performance of DeepECtransformer has suggested a list of entries that need careful inspection of whether they have mis-annotated EC numbers in the UniProt Knowledgebase (UniProtKB).” in lines 71-73 of revised manuscript.

8. The introduction section has no mention of existing EC prediction methods other than DeepEC. The authors need to highlight some existing methods along with their limitations (if any) to contrast with DeepECtransformer and/or any challenges that the current work specifically addresses. The authors should at least talk about some of the recent ones such as Ref. 2 and Ref. 3. Moreover, the authors themselves state in the manuscript (Lines 250-251), a recently developed method CLEAN showed improved performance compared to DeepECtransformer. Also, ProteInfer addresses several aspects that are not addressed in this study like no arbitrary sequence length cutoffs and the ability to predict more than just the EC number using GO labels. All this should be appropriately discussed.

[Response] Thank you for the comments. We agree. To address the reviewer’s comment and to provide a more specific improvement of DeepECtransformer in the aspect of interpretation of AI, we introduced ProteInfer, HDMLF, and CLEAN in the Introduction section, lines 56-65, as follows: “Various deep learning models for the prediction of EC numbers have also been developed. For instance, HDMLF was developed by integrating multiple sequence alignment with a deep neural network leveraging learned representations from a protein language model and bidirectional gated recurrent units²³. CLEAN, another deep learning model addressed imbalances in EC number distribution within the training dataset by employing contrastive learning, leading to prediction performance superior to the previous models²⁴. However, these models did not provide insights into the interpretability of AI reasoning. ProteInfer used a deep dilated convolutional network for EC number prediction and also provided interpretation of the prediction by class activation mapping²⁵. Nevertheless, the class activation mapping yielded coarse-grained feature maps, lacking fine-grained details, which are important for the residue-level analysis of protein sequences.”. To provide a comprehensive perspective on deep learning-based protein function predictions, we also included a discussion of aspects not addressed in this study in lines 311-318: “While DeepECtransformer has an advantage in terms of interpretability by providing attention weight-based fine-grained details, it is important to acknowledge that other EC number prediction tools address different facets of the problem. For instance, CLEAN tackles class imbalance by utilizing contrastive learning rather than supervised learning. ProteInfer uses a deep dilated convolutional network, which is not constrained by input amino acid sequence length and extends its predictions to include Gene Ontology (GO) terms, thereby offering a richer information of protein functionality.”.

9. Line 82 needs further explanation. How is the increase in coverage from 2,802 to 5,360 arrived at?

[Response] Thank you. We counted the total number of EC numbers that DeepECtransformer can predict by considering both EC numbers covered by the neural network and homology search. We elaborated on the manuscript as follows: “Including EC numbers that can be predicted by neural network and homology search, DeepECtransformer covers a total of 5,360 EC numbers.”.

10. Extended Data Fig. 1 lacks clear description. How do the authors arrive at a distribution of number of sequences for each EC class (these should be single numbers?) ?

[Response] Thank you. We counted the number of available sequences for each EC number (e.g., there are 52,567 sequences for EC:1.1.1.100 in the dataset) and visualized the distribution densities of the number of sequences for EC numbers as a density plot (Extended Data Fig. 1). EC numbers that belong to EC:1 tend to have a smaller number of sequences in the dataset compared to the number of sequences for other EC numbers. To clearly describe the analysis, we rewrote the lines 102-108 as follows: “The low performance for EC:1 class resulted from the inherent dataset imbalance, as the EC:1 class exhibited the lowest average number of sequences per EC number, with an average of 4,352 sequences compared to the other EC number classes (ranging from 6,819 sequences for EC:3 to 16,525 sequences for EC:6). Additionally, a statistical analysis of the data distribution confirmed that EC numbers belonging to EC:1 class generally had fewer sequences compared to other EC number classes (one-way ANOVA test, p-value < 0.001) (Extended Data Fig. 1).”. In addition, we added y-axis labels to the Extended Data Fig. 1, and rewrote the legend as follows: “The density plots of the number of sequences for EC number classes in the uniprot dataset. Each individual plot represents the density of the number of sequences for EC numbers that have the same first digit of the EC number.”.

References

1. Radivojac, P., Clark, W., Oron, T. et al. A large-scale evaluation of computational protein function prediction. *Nat Methods* 10, 221–227 (2013)
2. Sanderson, T., Bileschi, M. L., Belanger and D., Colwell, L. J., ProteInfer, deep neural networks for protein functional inference. *eLife* 12:e80942 (2023)
3. Yu, T., Cui, H., Li, J. C., et al., Enzyme function prediction using contrastive learning. *Science* 379,1358-1363 (2023)

Reviewer #2 (Remarks to the Author):

In the work, Kim et al. developed a deep learning model to predict the EC number for enzymes, which were then applied to predict the function for the unannotated genes in *E. coli*. Altogether,

the model can then be used to identify mis-annotated enzymes in database, identify functional regions in the enzyme, and also identify emerged functions during the evolution, making it a tool with broad applications in the biology field. Overall, the manuscript is well structured and organized, addressing relevant scientific questions enzyme function prediction. However, I have a few comments that might be helpful to improve the study.

[Response] Thank you for your time and effort of reviewing our manuscript, and providing constructive comments. We have addressed your comments as shown below.

1. The comparison between DeepECtransformer and the recently published CLEAN (Science 379, 1358–1363, 2023) doesn't appear to be sufficient. If the author claim that DeepECtransformer's capability to predict EC numbers for unannotated genes surpasses that of CLEAN, it is necessary to compare how many proteins with EC numbers can be predicted by CLEAN, in terms of the y-tome (1600 genes) mentioned by the author.

[Response] Thank you for the comments. CLEAN employs contrastive learning to predict EC numbers for a given protein sequence, embedding protein sequences into a latent space. Within this space, protein sequences with known EC numbers are embedded and subsequently clustered based on their EC numbers. Then, CLEAN identifies enzyme clusters that are most closely related to the query protein. Although the training scheme enables high prediction performance, it has a limitation in that it assigns EC numbers even to non-enzyme sequences. Consequently, CLEAN has predicted EC numbers for all y-ome proteins, which may contain non-enzyme proteins. This can lead to unintended false positives, particularly when analyzing protein functions in newly sequenced organisms. According to the comment, we discussed the prediction results of CLEAN for the y-ome proteins in lines 304-311 as follows: “Also, the use of CLEAN for the annotation of uncharacterized proteins requires careful interpretation of the prediction results because CLEAN assigns EC numbers for any input amino acid sequences, including non-enzyme amino acid sequences. CLEAN provides the confidence level (i.e., high, medium, low) of the predictions using a Gaussian mixture model. However, as the confidence level does not provide a detailed interpretation of how AI performs the reasoning process, careful inspection of the predictions should be conducted for the analysis of uncharacterized proteins, especially when the uncharacterized proteins contain non-enzyme proteins.”.

2. Additionally, the author's comparison of DeepECtransformer with other methods is limited to the Price-192 data, omitting the use of NEW-392 and the newly generated in-house data from CLEAN for comparison. Although the author mentioned that NEW-392 data was employed for HDMLF training, it remains essential to present a comparison between DeepECtransformer and CLEAN, as well as the second-best model ProteInfer (as indicated in the CLEAN paper in terms of this dataset). Furthermore, the second-best model BlastP, with precision of 0.5083, recall of 0.375, and an F1 score of 0.385 (as outlined in the CLEAN paper), also appears to outperform DeepECtransformer in terms of Price-192 data. This aspect should also be acknowledged, compared, or elucidated.

[Response] Thank you for the comments. According to your comments, we added the comparison of the prediction performance of EC number predictions tools (DeepECtransformer, CLEAN, and ProteInfer) on the NEW-392 dataset, which is presented in Supplementary Table 3. We also evaluated the prediction performance of BLASTp on the Price-149 dataset employing the parameters described in the CLEAN paper (i.e., using non-redundant UniProt/Swiss-Prot database as the subject database and BLOSUM62 matrix, assigned the EC number of a sequence having the top score). However, we were not able to reproduce the prediction performance described in the CLEAN paper. The prediction result is included in Supplementary Table 2. We also added discussions on the advantage of using deep learning-based tools over BLASTp in Supplementary Note 3 as follows: “While BLASTp exhibited comparable prediction performance to deep learning-based EC number prediction tools, the utilization of graphics processing unit (GPU)-based acceleration allows deep learning-based tools to achieve high-throughput prediction of EC numbers.”

3. As mentioned in Line 421, there are some EC number not covered by DeepECtransformer, please list the reason and also compare the EC number coverage with CLEAN.

[Response] Thank you for the comment. DeepECtransformer can predict a total of 5,360 EC numbers, which include those covered by the neural network as well as those covered by homology search. As we evaluated the performance on the Price-149 dataset and NEW-392 dataset, instead of the NEW-Price dataset that was mentioned in the line 421, in the revised manuscript, we removed the description (the mentioned line 421) for the dataset processing. Regarding the coverage of DeepECtransformer and CLEAN, DeepECtransformer and CLEAN cover 5,360 EC numbers and 5,242 EC numbers, respectively. Also, there are 4,894 EC numbers that are predictable by both DeepECtransformer and CLEAN. We added a Venn diagram that describes the coverage of DeepECtransformer and CLEAN in Supplementary Fig. 2.

4. Line 214-215: an explanation for the rationale behind selecting only three genes for experimental validation is notably absent. Clarification regarding the basis for this particular choice would enhance the understanding of the experimental design.

[Response] Thank you for the valuable comment. We aimed to show that DeepECtransformer can predict EC numbers for uncharacterized proteins. We first selected three representative EC number classes for experimental validation: oxidoreductase (EC:1), transferase (EC:2), and hydrolase (EC:3). To validate the predicted enzyme function through *in vitro* enzyme assays, we selected 179 proteins from 254 proteins (which are oxidoreductase, transferase, and hydrolase) that are predicted to be soluble by NetSolP [PMID: 35088833]. Among the 179 proteins, we randomly selected three targets (i.e., YgfF, YciO, and YjdM) each from EC:1, EC:2, and EC:3. To elaborate on the processing step, we added the following sentences: “To validate the ability of DeepECtransformer in identifying metabolic functions that cannot be detected by DeepEC and Swiss-Prot functional annotation processes, we performed *in vitro* enzyme assays to validate the predicted enzyme functions. We first selected three representative EC number classes, namely

oxidoreductase (EC:1), transferase (EC:2), and hydrolase (EC:3) to show the capability of DeepECtransformer to predict unknown enzyme functions. Among the 295 proteins that are exclusively predicted by DeepECtransformer to have all four digits of EC numbers, 179 proteins are predicted to be soluble in *E. coli* by NetSolP, a deep learning model for protein solubility prediction (Supplementary Data 7)³¹. From the 179 proteins, we randomly selected three proteins, YgfF, YciO, and YdjM, that are predicted to be oxidoreductase, transferase, and hydrolase, respectively.” in lines 233-241. Also, we made the prediction result of NetSolP for the 295 candidate proteins as the Supplementary Data 7.

5. It would be valuable to provide a justification for the chosen data split method. The current approach involves randomly dividing the dataset into training, validation, and test sets at an 8:1:1 ratio. However, considering the presence of numerous highly similar sequences within the UniProt database, there is a potential concern that this random split might inadvertently allocate highly similar sequences to both the training and test sets. If the objective is to develop a predictive model capable of generalizing effectively to data points that exhibit dissimilarity from the training data, it could be advisable to consider adopting a data split methodology akin to that employed in the CLEAN study. Alternatively, the author might consider evaluating sequence similarities between the training and test datasets.

[Response] Thank you for the constructive comments. For the training of a multi-class classifier by supervised learning, it is essential that all classes are represented in all of the training, validation and test datasets, to ensure a comprehensive assessment of performance. However, the division of the dataset based on the similarity-based clustering can lead to the omission of certain EC numbers in the validation or test datasets, primarily due to inherent dataset imbalances. Therefore, we randomly divided the dataset, and confirmed that all of the EC numbers are shared among the training, validation and test datasets. To address the concern that the reviewer pointed out, we calculated the sequence similarity between the entire training dataset and test dataset and re-evaluated the prediction performance of EC number prediction tools as the reviewer suggested (Extended Data Fig. 3). We confirmed that DeepECtransformer still shows improved ability to predict EC numbers for enzymes with relatively low sequence similarities to those in the training dataset.

6. One would expect that the EC class with more datapoints should achieve better performance in terms of prediction. Could the authors discuss this with results shown in Figure 2?

[Response] Thank you for the comment. As the reviewer pointed out, the performance of the neural network for an EC number is correlated with the number of datapoints for the EC number. We have conducted an analysis confirming a positive correlation between the number of sequences per EC number and the F₁ score for the EC number prediction (Spearman coefficient of 0.6872, $p < 1e-3$) in lines 109-111. While Figure 2 provides insights into the interpretation of the DeepECtransformer neural network, focusing on its interpretative capabilities rather than prediction performance, we have addressed the outcome of this analysis in Figure 1e.

7. How does the authors handle enzymes with multiple EC numbers during the training and prediction?

[Response] Thank you for the comment. The neural network was trained to handle a multi-class and multi-label prediction task. In contrast to conventional multi-class and single-label classifiers that typically employ a softmax layer as the last layer of the network, we used a sigmoid layer to calculate the output score of the prediction within the range of 0 to 1. To determine the threshold distinguishing positive and negative predictions for each EC number, we identified the threshold that maximized the F₁ score of the prediction performance in the validation dataset. To elaborate on the prediction process of multiple EC numbers, we replaced the following sentence in lines 463-466 “For the inference, we optimized thresholds of the positive and negative predictions.” to “For the inference, an EC number is assigned to the input sequence if the output score, calculated from the sigmoid layer, exceeds a threshold. In the case of promiscuous enzymes, all EC numbers exceeding their respective thresholds were assigned as the predicted EC numbers.”.

8. May the author also discuss the limitation of this method?

[Response] Thank you for the comment. We added the following sentences in lines 311-318 to discuss the limitation of DeepECtransformer: “While DeepECtransformer has an advantage in terms of interpretability by providing attention weight-based fine-grained details, it is important to acknowledge that other EC number prediction tools address different facets of the problem. For instance, CLEAN tackles class imbalance by utilizing contrastive learning rather than supervised learning. ProteInfer uses a deep dilated convolutional network, which is not constrained by input amino acid sequence length and extends its predictions to include Gene Ontology (GO) terms, thereby offering a richer information of protein functionality.”.

Minor:

1. Line 420: Please add explanation for “NEW-392 dataset” and “Price-149 dataset”.

[Response] Thank you. We added the explanation for the datasets as follows: “To compare the performance of EC number prediction tools, we used NEW-392 dataset, which contains 392 amino acid sequences covering 177 types of EC numbers, and Price-149 dataset, which contains 149 amino acid sequences covering 56 types of EC numbers, provided by Yu et al³⁸.”.

2. In the extended Data Fig.2, the comparison with CLEAN should also be added.

[Response] Thank you for the comment. We added the performance of CLEAN in the Extended Data Fig. 2.

3. Line 208: Please explain the reason for 74 proteins (464 protein – 390 proteins) with predicted EC numbers but does not contain four full digits.

[Response] Thank you for the comment. DeepECtransformer employs both neural network and homology search to predict EC numbers. Among the 5,360 EC numbers that are covered by

DeepECtransformer, 218 EC numbers have incomplete EC numbers, which are predicted by homology search. The excluded 74 proteins were predicted to have the incomplete EC numbers. We added a Venn diagram that describes the coverage of DeepECtransformer and CLEAN in Supplementary Figure 2.

4. Line 220, YciO was previously be annotated as the Suc5 family protein by other algorithms as shown in UniProt database. If so, please mention this in the text.

[Response] Thank you. Accordingly, we mentioned that YciO was previously annotated to belong to the SUA5 family.

Reviewer #3 (Remarks to the Author):

This article introduces a neural network model, DeepECtransformer, to predict EC numbers of enzymes based on the amino acid sequences. The technique looks promising and will help the research community predict metabolic functions of organisms using just sequence information.

[Response] Thank you for your time and effort of reviewing our manuscript, and providing constructive comments. We addressed your comments as shown below.

Overall, the extent of advance over their prior approach, deepEC, seems limited. The rationale and advantage of using transformer layers is never discussed. Limitations of this new strategy are also not discussed. What were the sources of errors with the original deepEC and in what cases does this new approach improve upon it? How exactly were the limitations of deepEC resolved with this new approach?

[Response] Thank you for the great comments. Previous DeepEC neural network was trained on a dataset comprising 1,360,727 amino acid sequences covering 2,240 EC numbers. In this study, DeepECtransformer shows much improved prediction performance and EC number coverage by training on a more extensive dataset, which includes 22,477,695 amino acid sequences covering 2,802 EC numbers. These improvements in prediction performance was analyzed in Table 1, Extended Data Fig. 2 and Extended Data Fig. 3. We have further experimentally validated that DeepECtransformer was able to predict EC numbers of uncharacterized γ -ome proteins (i.e., Ygff, YciO, and YjdM), which were not predicted by previous DeepEC. Regarding the

advantage of using transformer layers, we have harnessed their interpretative potential to unveil the black boxes of the neural network using the attention scores calculated in the transformer layers. To provide a more specific improvement of DeepECtransformer in the aspect of interpretation of AI, we added the following sentences in lines 56-65: “Various deep learning models for the prediction of EC numbers have also been developed. For instance, HDMLF was developed by integrating multiple sequence alignment with a deep neural network leveraging learned representations from a protein language model and bidirectional gated recurrent units²³. CLEAN, another deep learning model addressed imbalances in EC number distribution within the training dataset by employing contrastive learning, leading to prediction performance superior to the previous models²⁴. However, these models did not provide insights into the interpretability of AI reasoning. ProteInfer used a deep dilated convolutional network for EC number prediction and also provided interpretation of the prediction by class activation mapping²⁵. Nevertheless, the class activation mapping yielded coarse-grained feature maps, lacking fine-grained details, which are important for the residue-level analysis of protein sequences.”. Also, to provide more clear advantage of using transformer layers in the neural network, we replaced the sentence in lines 74-76 of the revised manuscript, “By analyzing the regions of focus during the prediction of enzyme functions by AI, we have confirmed that DeepECtransformer has learned to identify important regions, such as active sites or cofactor binding sites.” to “By analyzing the regions of focus during the prediction of enzyme functions by transformer layers, we have confirmed that DeepECtransformer has learned to identify important regions, such as active sites or cofactor binding sites.”. To address the comment regarding the limitation of DeepECtransformer, we added the following sentences in lines 311-318 to further discuss the limitation of DeepECtransformer: “While DeepECtransformer has an advantage in terms of interpretability by providing attention weight-based fine-grained details, it is important to acknowledge that other EC number prediction tools address different facets of the problem. For instance, CLEAN tackles class imbalance by utilizing contrastive learning rather than supervised learning. ProteInfer uses a deep dilated convolutional network, which is not constrained by input amino acid sequence length and extends its predictions to include Gene Ontology (GO) terms, thereby offering a richer information of protein functionality.”.

Also a new approach, called CLEAN, appears to outperform in every category. Why was CLEAN not able to make predictions for the two y-genes? Were there cases where CLEAN made predictions and their approach didn't? This field is still developing, and new tools are needed. CLEAN and DeepEC are still work in progress, but it would help to clarify strengths/weaknesses of these methods. CLEAN should be reported in the main text and figures too.

[Response] Thank you for pointing out the differences in the applicability of DeepECtransformer and CLEAN. CLEAN employs contrastive learning to predict EC numbers for a given protein sequence by embedding the sequence into a latent space. Within this space, protein sequences with known EC numbers are embedded and subsequently clustered based on their EC numbers.

Then, CLEAN identifies enzyme clusters that are most closely related to the query protein. Although the training scheme enables high prediction performance, it has a limitation in that it assigns EC numbers even to non-enzyme sequences. Consequently, CLEAN has predicted EC numbers for all y-ome proteins, which may contain non-enzyme proteins. This can lead to unintended false positives, particularly when analyzing protein functions in newly sequenced organisms. According to the comment, we discussed the prediction result of CLEAN for the y-ome proteins in lines 304-311 as follows: “Also the use of CLEAN for the annotation of uncharacterized proteins requires careful interpretation of the prediction results because CLEAN assigns EC numbers for any input amino acid sequences, including non-enzyme amino acid sequences. CLEAN provides the confidence level (i.e., high, medium, low) of the predictions using a Gaussian mixture model. However, as the confidence level does not provide a detailed interpretation of how the AI makes the reasoning process, careful inspection of the predictions should be conducted for the analysis of uncharacterized proteins, especially when the uncharacterized proteins contain non-enzyme proteins.”. We also added the following sentences in the Introduction section to discuss the differences in the interpretability of CLEAN and DeepECTransformer: “Various deep learning models for the prediction of EC numbers have also been developed. For instance, HDMLF was developed by integrating multiple sequence alignment with a deep neural network leveraging learned representations from a protein language model and bidirectional gated recurrent units²³. CLEAN, another deep learning model addressed imbalances in EC number distribution within the training dataset by employing contrastive learning, leading to prediction performance superior to the previous models²⁴. However, these models did not provide insights into the interpretability of AI reasoning. ProteInfer used a deep dilated convolutional network for EC number prediction and also provided interpretation of the prediction by class activation mapping²⁵. Nevertheless, the class activation mapping yielded coarse-grained feature maps, lacking fine-grained details, which are important for the residue-level analysis of protein sequences.”. To further discuss the prediction performance of CLEAN in the main text, we also added the analysis of prediction performance of CLEAN in Extended Data Fig. 2 and 3.

How are enzymes with multiple activities (ECs) handled in training/testing and novel predictions?

[Response] Thank you for the comment. The neural network was trained to handle a multi-class and multi-label prediction task. In contrast to conventional multi-class and single-label classifiers that typically employ a softmax layer as the last layer of the network, we used a sigmoid layer to calculate the output score of the prediction within the range of 0 to 1. To determine the threshold distinguishing positive and negative predictions for each EC number, we identified the threshold that maximized the F₁ score of the prediction performance in the validation dataset. To elaborate on the prediction process of multiple EC numbers, we replaced the following sentence in lines 463-466 “For the inference, we optimized thresholds of the positive and negative predictions.” to “For the inference, an EC number is assigned to the input sequence if the output score, calculated

from the sigmoid layer, exceeds a threshold. In the case of promiscuous enzymes, all EC numbers exceeding their respective thresholds were assigned as the predicted EC numbers.”.

There’s a cryptic statement in the results, which is not described in detail, but seems very important in influencing its accuracy -

“If the neural network fails to predict EC numbers, homologous enzymes are analyzed and EC numbers are assigned using UniProtKB/Swiss-Prot enzymes as the subject database”.

For how many cases did it fail? What scenarios does it commonly fail? What’s the accuracy without the homology approach? Are failures considered as incorrect predictions?

[Response] Thank you for the comment. The homology analysis is performed when the neural network of DeepECtransformer predicts no EC numbers for an input amino acid sequence. As the neural network was trained to predict EC numbers within the 2,802 EC numbers, if an input amino acid sequence has different functions that are covered by the 2,802 EC numbers, the neural network will predict no EC numbers. The prediction performance without the homology analysis, which is the prediction performance of the neural network, was evaluated in Fig. 1. As the term “fail” was confusing, we replaced the sentence to “If the neural network predicts no EC numbers for a given amino acid sequence, homologous enzymes for the amino acid sequence are analyzed using UniProtKB/Swiss-Prot enzymes as the subject database and EC numbers of the homologous enzymes are assigned^{4, 28}”.

Choice of test set: the authors randomly sampled 10% of the entire data. Issues of data leakage are not addressed when selecting an appropriate test set. What is the accuracy when there isn’t a close homolog in the training? What’s the accuracy in annotating a ‘new’ organism or a ‘new’ enzyme class?

[Response] Thank you for the comments. To address the concern, we calculated the sequence similarity between the entire training dataset and test dataset and re-evaluated the prediction performance of EC number prediction tools. The analysis result is included in Extended Data Fig. 3. We confirmed that DeepECtransformer still shows improved ability to predict EC numbers for enzymes with relatively low sequence similarities to those in the training dataset.

How were ‘misannotated’ enzymes chosen for experimental testing? Currently, they seem to have arbitrarily picked 3 enzymes.

[Response] In the section “Discovering the unknown functions of enzymes in *E. coli* K-12 MG1655”, we tried to show that DeepECtransformer can predict diverse EC number classes. Therefore, we first chose three representative EC number classes to validate by the experiment, which were oxidoreductase (EC:1), transferase (EC:2), and hydrolase (EC:3). To validate the predicted enzyme function using *in vitro* enzyme assay, we selected 179 proteins from 254 proteins (which are oxidoreductase, transferase, and hydrolase) that are predicted to be soluble by NetSolP [PMID: 35088833]. Among the 179 proteins, we randomly selected three targets (i.e., YgfF, YciO, and YjdM) each from EC:1, EC:2, and EC:3. To elaborate the processing step,

we added the following sentences: “To validate the ability of DeepECtransformer in identifying metabolic functions that cannot be detected by DeepEC and Swiss-Prot functional annotation processes, we performed *in vitro* enzyme assays to validate the predicted enzyme functions. We first selected three representative EC number classes, namely oxidoreductase (EC:1), transferase (EC:2), and hydrolase (EC:3) to show the capability of DeepECtransformer to predict unknown enzyme functions. Among the 295 proteins that are exclusively predicted by DeepECtransformer to have all four digits of EC numbers, 179 proteins are predicted to be soluble in *E. coli* by NetSolP, a deep learning model for protein solubility prediction (Supplementary Data 7)³¹. From the 179 proteins, we randomly selected three proteins, YgfF, YciO, and YdjM, that are predicted to be oxidoreductase, transferase, and hydrolase, respectively.”. Also, we made the prediction result of NetSolP for the 295 candidate proteins as the Supplementary Data 7.

Minor comments

Line 109, they identified correct activities for some ‘misannotated’ enzymes; while they have shown proof for the new activity, they didn’t show evidence that the original activity was incorrect. Maybe they are multifunctional/promiscuous enzymes? How was the original activity determined?

[Response] Thank you for the comment. To confirm that P93052 functions as a malate dehydrogenase rather than as an L-lactate dehydrogenase, we expressed *P93052* gene in the *Mannheimia succiniciproducens* PALK strain which was constructed with deletion of L-lactate dehydrogenase gene. While fed-batch fermentation of wild-type *M. succiniciproducens* produces lactic acid, the fed-batch fermentation of the PALK strain produces no lactic acid. When we conducted fed-batch fermentation with PALK (pMS3-P93052), expressing *P93052* gene in PALK strain, we observed no accumulation of lactic acid. These results suggest that P93052 does not have any enzymatic activity related to L-lactate dehydrogenase. The detailed experimental results are included in Supplementary Note 1.

Line 179-197, related to *E. coli* strains, is there any evidence supporting these predictions?

[Response] Thank you for the comment. In the section, we found that there are only six *E. coli* strains (i.e., *E. coli* KTE11, KTE31, KTE33, KTE96, KTE114, and KTE159) among the 1,122 strains that possess the three AroL which are predicted to have additional EC number (EC:1.1.1.159). The six strains exclusively possess an allele *hdhA_61* (which encodes 7-alpha-hydroxysteroid dehydrogenase; EC:1.1.1.159) while the other 1,116 strain possess different *hdhA* alleles, further supporting that the six strains have experienced distinct evolution with the other strains. In a recent study, the six strains were identified as belonging to the same phylogroup of *E. coli* strains (U/cryptic phylogroup), further supporting that the strains have experienced a common evolutionary trajectory [PMID: 37011218]. Accordingly, we added the sentence “In a recent study, these six strains were also identified as belonging to the same phylogroup of *E. coli* strains³¹.” in lines 204-205 of the revised manuscript.

Line 67: Authors mentioned that DeepECtransformer predicted 464 proteins out of 1569, why is that so? Why only 464? Is there a case where this model will fail in predicting the EC number (doesn't the homology approach handle that (line 80-81))?

[Response] Thank you for the comment. The 1,569 protein sequences for the y-ome genes also contain non-enzyme sequences, which would not have corresponding EC numbers. Therefore, DeepECtransformer, including both the neural network engine and the homology search engine, did not predict EC numbers for the remaining 1,105 proteins.

Line 88: Inconsistent caption of the figure (1d) and description in text

[Response] Thank you for the comment. Both the line 88 and the caption of the Fig. 1d describe the distribution of EC numbers (e.g., there are 720 EC number classes that belong to the oxidoreductase; EC:1). To clearly describe the figure, we replaced the caption of the Fig. 1d from "Distribution of EC numbers per first-level EC number." to "Distribution of the types of EC numbers per first-level EC number."

Lines 110-114: Authors talk about mis annotations of proteins, Q8U4R3 and Q038Z3, and claim that their approach has accurately predicted it. Do the authors have any proof of the validity and correctness of their claim? Is there a confidence score associated with these predictions?

[Response] Thank you for the valuable comments. To provide insights into the prediction, we added explanations of the attention weights of the example proteins and sequence similarity between the training dataset and the example proteins in Supplementary Note 2. and cited it in the manuscript lines 140-143 to elaborate on the predictions as follows: "As the EC numbers suggested by DeepECtransformer are predictions, we further analyzed how the AI made the predictions *in silico* (Supplementary Note 2). Even though *in silico* analysis of the predictions can provide clues for the predicted functionality, it is necessary to experimentally validate their functions."

Line 208/215: Authors mention 390 proteins have full EC numbers, while later they mention 295 proteins have full EC numbers.

[Response] Thank you for the comment. As described in lines 229-233 in the revised manuscript, we excluded the proteins whose EC numbers are also predicted by DeepEC or are annotated in the UniProt database to show the advantage of DeepECtransformer. To clearly convey the number of proteins being analyzed, we added the following sentence in lines 231-232 of the revised manuscript: "DeepECtransformer exclusively predicted complete four-digit EC numbers for 295 proteins."

Lines 330-365 in methods, I'm unsure why it's there, this organism and bioreactor setup was not discussed in results.

[Response] Thank you for the comment. The analysis was performed to confirm the malate dehydrogenase activity of P93052, which has been discussed in lines 122-126 of the revised

manuscript and Supplementary Note 1. The methodologies for the fed-batch fermentation have been relocated to Supplementary Note 4.

Table 1 Define micro and macroprecision/recall/f1 in the Methods section.

[Response] Thank you. We added the definition of the evaluation metrics in the Methods section as follows.

“The evaluation metrics are calculated as follows. *Macro precision* = $\frac{1}{C} \sum_{i=1}^C \frac{TP_i}{TP_i+FP_i}$,

$$\text{macro recall} = \frac{1}{C} \sum_{i=1}^C \frac{TP_i}{TP_i+FN_i}, \text{macro } F_1 \text{ score} = \frac{1}{C} \sum_{i=1}^C \frac{2 \cdot \text{Precision}_i \cdot \text{Recall}_i}{\text{Precision}_i + \text{Recall}_i},$$

$$\text{micro precision} = \frac{\sum_{i=1}^C TP_i}{\sum_{i=1}^C (TP_i+FP_i)}, \text{micro recall} = \frac{\sum_{i=1}^C TP_i}{\sum_{i=1}^C (TP_i+FN_i)}, \text{micro } F_1 \text{ score} =$$

$\frac{2 \cdot \text{micro precision} \cdot \text{micro recall}}{\text{micro precision} + \text{micro recall}}$, where TP_i is the number of true positive prediction for class i , FP_i is the number of false positive prediction for class i , FN_i is the number of false negative prediction for class i , and C is the number of classes.”

Figure 1 – include deepEC architecture for context.

[Response] Thank you for the comment. We agree that the comparison of neural network architectures between DeepEC and DeepECtransformer can clearly explain the updates in this manuscript. However, as the neural network architecture of DeepEC is discussed as a supplementary information that describes the differences between the model, we added a Figure of the DeepEC neural network architecture as Supplementary Fig. 1.

Fig 4 – include the specific activity tested (i.e. substrate/product)

[Response] Thank you. We added reaction equations with the molecular structure of the substrate and product in Figure 4.

Ex Fig 1 – what is the y axis? Needs a more detailed caption.

[Response] Thank you. The y-axis represents the density of the number of sequences for EC number classes. To clearly describe the analysis, we rewrote the lines 102-108 of the revised manuscript as follows: “The low performance for EC:1 class resulted from the inherent dataset imbalance, as the EC:1 class exhibited the lowest average number of sequences per EC number, with an average of 4,352 sequences compared to the other EC number classes (ranging from 6,819 sequences for EC:3 to 16,525 sequences for EC:6). Additionally, a statistical analysis of the data distribution confirmed that EC numbers belonging to EC:1 class generally had fewer sequences compared to other EC number classes (one-way ANOVA test, p-value < 0.001) (Extended Data Fig. 1).”. In addition, we added y-axis labels to the Extended Data Fig. 1, and rewrote the legend as follows: “The density plots of the distribution of the number of sequences for EC number classes in the uniprot dataset. Each individual plot represents the density of the number of sequences for EC numbers that have the same first-level EC number.”.

Ex Fig 9 – where is this analysis used? A longer caption is needed.

[Response] Thank you for the comment. The analysis was cited in the Methods section (Homologous enzyme searches) and detailed information was also described in the section. To elaborate on the legend of Extended Data Fig. 9, we added the following sentences to the legend: “A parameter set with a minimum sequence identity of 50% and a minimum sequence coverage of 75% showed the highest micro F_1 score was selected as the hyperparameter set used in this study.”.

Reviewer #1 (Remarks to the Author):

The authors addressed all our points requiring classification.

Reviewer #2 (Remarks to the Author):

The authors have answered all my comments.

Reviewer #3 (Remarks to the Author):

the authors have adequately addressed my concerns